# Adaptive Client Sampling in Federated Learning via Online Learning with Bandit Feedback

## Abstract

Due to the high cost of communication, federated learning (FL) systems need to sample a subset of clients that are involved in each round of training. As a result, client sampling plays an important role in FL systems as it affects the convergence rate of optimization algorithms used to train machine learning models. Despite its importance, there is limited work on how to sample clients effectively. In this paper, we cast client sampling as an online learning task with bandit feedback, which we solve with an online stochastic mirror descent (OSMD) algorithm designed to minimize the sampling variance. We then theoretically show how our sampling method can improve the convergence speed of optimization algorithms. To handle the tuning parameters in OSMD that depend on the unknown problem parameters, we use the online ensemble method and doubling trick. We prove a dynamic regret bound relative to any sampling sequence. The regret bound depends on the total variation of the comparator sequence, which naturally captures the intrinsic difficulty of the problem. To the best of our knowledge, these theoretical contributions are new and the proof technique is of independent interest. Through both synthetic and real data experiments, we illustrate advantages of the proposed client sampling algorithm over the widely used uniform sampling and existing online learning based sampling strategies. The proposed adaptive sampling procedure is applicable beyond the FL problem studied here and can be used to improve the performance of stochastic optimization procedures such as stochastic gradient descent and stochastic coordinate descent.

## 1 Introduction

Modern edge devices, such as personal mobile phones, wearable devices, and sensor systems in vehicles, collect large amounts of data that are valuable for training of machine learning models. If each device only uses its local data to train a model, the resulting generalization performance will be limited due to the number of available samples on each device. Traditional approaches where data are transferred to a central server, which trains a model based on all available data, have fallen out of fashion due to privacy concerns and high communication costs. Federated Learning (FL) has emerged as a paradigm that allows for collaboration between different devices (clients) to train a global model while keeping data locally and only exchanging model updates (McMahan et al., 2017).

In a typical FL process, we have clients that contain data and a central server that orchestrates the training process (Kairouz et al., 2021). The following process is repeated until the model is trained: (i) the server selects a subset of available clients; (ii) the server broadcasts the current model parameters and sometimes also a training program (e.g., a Tensorflow graph (Abadi et al., 2016)); (iii) the selected clients make updates to the model parameters based on their local data; (iv) the local model updates are uploaded to the server; (v) the server aggregates the local updates and makes a global update of the shared model. In this paper, we focus on the first step and develop a practical strategy for selecting clients with provable guarantees.

To train a machine learning model in a FL setting with $M$ clients, we would like to minimize the following objective:[1]

$$\min_{w} F(w) := \sum_{m \in [M]} \lambda_m \phi\left(w; \mathcal{D}_m\right),\tag{1}$$

where $\phi(w; \mathcal{D}_m)$ is the loss function used to assess the quality of a machine learning model parameterized by the vector $w$ based on the local data $\mathcal{D}_m$ on the client $m \in [M]$. The parameter $\lambda_m$ denotes the weight for client $m$. Typically, we have $\lambda_m = n_m/n$, where $n_m = |\mathcal{D}_m|$ is the number of samples on the client $m$, and the total number of samples is $n = \sum_{m=1}^{M} n_m$. At the beginning of the $t$-th communication round, the server uses the sampling distribution $p^t = (p_1^t, \ldots, p_M^t)^\top$ to choose $K$ clients by sampling with replacement from $[M]$.[2] Let $S^t \subseteq [M]$ denote the set of chosen clients with $|S^t| = K$. The server transmits the current model parameter vector $w^t$ to each client $m \in S^t$. The client $m$ computes the local update $g_m^t$ and sends it back to the server.[3] After receiving local updates from clients in $S^t$, the server constructs a stochastic estimate of the global gradient as

$$g^t = \frac{1}{K} \sum_{m \in S^t} \frac{\lambda_m}{p_m^t} g_m^t,\tag{2}$$

and makes the global update of the parameter $w^t$ using $g^t$. For example, $w^{t+1} = w^t - \mu_t g^t$, if the server is using stochastic gradient descent (SGD) with the stepsize sequence $\{\mu_t\}_{t \geq 1}$ (Bottou et al., 2018). However, the global update can be obtained using other procedures as well.

The sampling distribution in FL is typically uniform over clients, $p^t = p^{\text{unif}} = (1/M, \ldots, 1/M)^\top$. However, nonuniform sampling (also called importance sampling) can lead to faster convergence, both in theory and practice, as has been illustrated in stochastic optimization (Zhao & Zhang, 2015; Needell et al., 2016). While the sampling distribution can be designed based on prior knowledge (Zhao & Zhang, 2015; Johnson & Guestrin, 2018; Needell et al., 2016; Stich et al., 2017), we cast the problem of choosing the sampling distribution as an online learning task and need no prior knowledge about equation 1.

Existing approaches to designing a sampling distribution using online learning focus on estimation of the best sampling distribution under the assumption that it does not change during the training process. However, the best sampling distribution changes with iterations during the training process, and the target stationary distribution does not capture the best sampling distribution in each round. In the existing literature, the best *fixed* distribution in hindsight is used as the comparator to measure the performance of the algorithm used to design the sampling distribution. Here, we focus on measuring the performance of the proposed algorithm against the best *dynamic* sampling distribution. We use an online stochastic mirror descent (OSMD) algorithm to generate a sequence of sampling distributions and prove a regret bound relative to any dynamic comparators that involve a total variation term that characterizes the intrinsic difficulty of the problem. To the best of our knowledge, this is the first bound on the dynamic regret with intrinsic difficulty characterization in importance sampling. Moreover, we theoretically show how our sampling method improves the convergence guarantee of optimization method by reducing the dependency on the heterogeneity of the problem.

## 1.1 CONTRIBUTIONS

We develop an algorithm based on OSMD that generates a sequence of sampling distributions $\{p^t\}_{t \geq 1}$ based on the partial feedback available to the server from the sampled clients. We prove a bound on regret relative to the any *dynamic* comparators, which allows us to consider the best sequence of sampling distributions as they change over iterations. The bound includes a *total variation*

---

[1]We use $[M]$ to denote the set $\{1, \ldots, M\}$.

[2]In this paper, we assume that all clients are available in each round and the purpose of client sampling is to reduce the communication cost, which is also the case considered by some previous research (Chen et al., 2020). However, in practice, it is possible that only a subset of clients are available at the beginning of each round due to physical constraint. In Appendix H.2, we discuss how to extend our proposed methods to deal with such situations. Analyzing such an extension is highly non-trivial and we leave it for further study. See detailed discussion in Appendix H.2.

[3]Throughout the paper we do not discuss how $g_m^t$ is obtained. One possibility that the reader could keep in mind for concreteness is the LocalUpdate algorithm Charles & Konečný (2020), which covers well-known algorithms such as mini-batch SGD and FedAvg (McMahan et al., 2017).

term that characterizes the *intrinsic difficulty* of the problem by capturing the difficulty of following the best sequence of distributions. Such a characterization of problem difficulty is novel. Besides, our theoretical result can recover the results in previous research as special cases and is thus strictly more general. Moreover, we theoretically improve the convergence guarantee of optimization algorithm by using our sampling scheme over uniform sampling. We show that adaptive sampling can help reduce the dependency on the heterogeneity level of the problem. We also mkae contributions in experiments and practical parameter tuning strategy. See Appendix A.1 for more detailed discussion of our contributions.

## 1.2 RELATED WORK

Our paper is related to client sampling in FL, importance sampling in stochastic optimization, and online convex optimization. We summarize only the most relevant literature, without attempting to provide an extensive survey. Due to space limit, we only summarize the first direction here, and leave the discussion about importance sampling in stochastic optimization and online convex optimization in Appendix B.

For client sampling, Chen et al. (2020) proposed to use the theoretically optimal sampling distribution to choose clients. However, their method requires all clients to compute local updates in each round, which is impractical due to stragglers. Ribero & Vikalo (2020) modelled the parameters of the model during training by an Ornstein-Uhlenbeck process, which was then used to derive an optimal sampling distribution. Cho et al. (2020b) developed a biased client selection strategy and analyzed its convergence property. As a result, the algorithm has a non-vanishing bias and is not guaranteed to converge to optimum. Moreover, it needs to involve more clients than our method and is thus communication and computational more expensive. Kim et al. (2020); Cho et al. (2020a); Yang et al. (2020) considered client sampling as a multi-armed bandit problem, but provided only limited theoretical results. Wang et al. (2020) used reinforcement learning for client sampling with the objective of maximizing accuracy, while minimizing the number of communication rounds.

## 1.3 NOTATION

Let $\mathbb{R}_+^M = [0, \infty)^M$ and $\mathbb{R}_{++}^M = (0, \infty)^M$. For $M \in \mathbb{N}^+$, let $\mathcal{P}_{M-1} := \{x \in \mathbb{R}^M : \sum_{i=1}^M x_i = 1\}$ be the $(M-1)$-dimensional probability simplex. We use $p = (p_1, \ldots, p_M)^\top$ to denote a sampling distribution with support on $[M] := \{1, \ldots, M\}$. We use $p^{1:T}$ to denote a sequence of sampling distributions $\{p^t\}_{t=1}^T$. Let $\Phi : \mathcal{D} \subseteq \mathbb{R}^M \mapsto \mathbb{R}$ be a differentiable convex function defined on $\mathcal{D}$, where $\mathcal{D}$ is a convex open set, and we use $\bar{\mathcal{D}}$ to denote the closure of $\mathcal{D}$. The Bregman divergence between any $x, y \in \mathcal{D}$ with respect to the function $\Phi$ is given as $D_\Phi(x \| y) = \Phi(x) - \Phi(y) - \langle \nabla \Phi(y), x - y \rangle$. The unnormalized negative entropy is denoted as $\Phi_e(x) = \sum_{m=1}^M x_m \log x_m - \sum_{m=1}^M x_m$, $x = (x_1, \ldots, x_M)^\top \in \mathcal{D} = \mathbb{R}_+^M$, with $0 \log 0$ defined as $0$. We use $\| \cdot \|_p$ to denote the $L_p$-norm for $1 \le p \le \infty$. For $x \in \mathbb{R}^n$, we have $\|x\|_p = (\sum_{i=1}^n x_i^p)^{1/p}$ when $1 < p < \infty$, $\|x\|_1 = \sum_{i=1}^n |x_i|$, and $\|x\|_\infty = \max_{1 \le i \le n} |x_i|$. Given any $L_p$-norm $\| \cdot \|$, we define its dual norm as $\|z\|_\star := \sup\{z^\top x : \|x\| \le 1\}$.

For two sequences $\{a_n\}$ and $\{b_n\}$, we use $a_n = O(b_n)$ or $a_n \lesssim b_n$ if there exists $C > 0$ such that $|a_n/b_n| \le C$ for all $n$ large enough; $a_n = \Theta(b_n)$ if $a_n = O(b_n)$ and $b_n = O(a_n)$ simultaneously. Similarly, $a_n = \tilde{O}(b_n)$ if $a_n = O(b_n \log^k b_n)$ for some $k \ge 0$; $a_n = \tilde{\Theta}(b_n)$ if $a_n = \tilde{O}(b_n)$ and $b_n = \tilde{O}(a_n)$ simultaneously.

## 1.4 ORGANIZATION OF THE PAPER

We motivate importance sampling in FL, introduce an adaptive client sampling algorithm, and establish a bound on the dynamic regret in Section 2. We derive optimization guarantee of mini-batch SGD using our sampling scheme in Section 3.

Due to space limit, we leave additional contents in appendix. More specifically, we give additional related work summary in Appendix B. In Appendix C.3 and Appendix C.4, we design two extensions to the sampling algorithm that make it adaptive to the unknown problem parameters. We leave detailed algorithm descriptions and theoretical properties of adaptive methods in Appendix C. We

provide the experimental results on synthetic data in Appendix F and real-world data in Appendix G. Sampling without replacement is discussed in Appendix C.5 and Appendix F.4. Finally, we give conclusions and discussions about future directions in Appendix H.

## 2 ADAPTIVE CLIENT SAMPLING

We show how to cast the client sampling problem as an online learning task. Subsequently, we solve the online learning problem using OSMD algorithm and provide a regret analysis for it.

### 2.1 CLIENT SAMPLING AS AN ONLINE LEARNING PROBLEM

Recall that at the beginning of the $t$-th communication round, the server uses a sampling distribution $p^t$ to choose a set of clients $S^t$, by sampling with replacement $K$ clients from $[M]$, to update the parameter vector $w^t$. For a chosen client $m \in S^t$, the local update is denoted as $g_m^t$. For example, the local update $g_m^t = \nabla \phi(w^t; \mathcal{D}_m)$ may be the full gradient; when mini-batch SGD/FedSGD is used, then $g_m^t = (1/B) \sum_{b=1}^B \nabla \phi(w^t; \xi_{m,b}^t)$, where $\xi_{m,b}^t \overset{i.i.d.}{\sim} \mathcal{D}_m$ and $B$ is the batch size; when FedAvg is used, then $g_m^t = (w_{m,B}^t - w^t)/\eta_{\text{local}}$, where $w_{m,b}^t = w_{m,b-1}^t - \eta_{\text{local}} \nabla f\left(w_{m,b-1}^t; \xi_{m,b-1}^t\right)$, $b \in [B]$, $w_{m,0}^t = w^t$, $\xi_{m,b}^t \overset{i.i.d.}{\sim} \mathcal{D}_m$, and $\eta_{\text{local}}$ is the local stepsize. We define the aggregated oracle update at the $t$-th communication round as

$$J^t = \sum_{m=1}^M \lambda_m g_m^t.$$

The oracle update $J^t$ is constructed only for theoretical purposes and is not computed in practice. The stochastic estimate $g^t$, defined in equation 2, is an unbiased estimate of $J^t$, that is, $\mathbb{E}_{S^t}[g^t] = J^t$. Note that we only consider the randomness of $S^t$ and treat $g_m^t$ as given.[4] The variance of $g^t$ is

$$\mathbb{V}_{S^t}[g^t] = \frac{1}{K} \left( \sum_{m=1}^M \frac{\lambda_m^2 \|g_m^t\|^2}{p_m^t} - \|J^t\|^2 \right). \tag{3}$$

Our goal is to design the sampling distribution $p^t$, used to sample $S^t$, so to minimize the variance in equation 3. In doing so, we can ignore the second term as it is independent of $p^t$.

Let $a_m^t = \lambda_m^2 \|g_m^t\|^2$. For any sampling distribution $q = (q_1, \ldots, q_M)^\top$, the *variance reduction loss*[5] is defined as

$$l_t(q) = \frac{1}{K} \sum_{m=1}^M \frac{a_m^t}{q_m}. \tag{4}$$

Given a sequence of sampling distributions $q^{1:T}$, the cumulative variance reduction loss is defined as $L(q^{1:T}) := \sum_{t=1}^T l_t(q^t)$. When the choice of $q^{1:T}$ is random, the expected cumulative variance reduction loss is defined as $\bar{L}(q^{1:T}) := \mathbb{E}[L(q^{1:T})]$.

The variance reduction loss appears in the bound on the sub-optimality of a stochastic optimization algorithm. As a motivating example, suppose $F(\cdot)$ in equation 1 is $\sigma$-strongly convex. Furthermore, suppose the local update $g_m^t = \nabla \phi(w^t; \mathcal{D}_m)$ is the full gradient of the local loss and the global update is made by SGD with stepsize $\mu_t = 2/(\sigma t)$. Theorem 3 of Salehi et al. (2017) then states that for any $T \geq 1$:

$$\mathbb{E}\left[ F\left( \frac{2}{T(T+1)} \sum_{t=1}^T t \cdot w^t \right) \right] - F(w^\star) \leq \frac{2}{\sigma T(T+1)} \bar{L}(p^{1:T}), \tag{5}$$

---

[4]The randomness comes from two sampling processes. The first sampling happens on clients level, and the second sampling happens locally when choosing samples. To ease the understanding, one may treat $g_m^t$ as full local gradient.

[5]The variance reduction loss $l_t(\cdot)$ should be distinguished from the training loss $\phi(\cdot)$. While the former is always convex, $\phi(\cdot)$ can be non-convex.

where $w^\star$ is the minimizer of the objective in equation 1. Therefore, by choosing the sequence of sampling distributions $p^{1:T}$ to make the $\bar{L}(p^{1:T})$ small, one can achieve faster convergence. This observation holds in other stochastic optimization problems as well. We develop an algorithm that creates a sequence of sampling distributions $p^{1:T}$ to minimize $\bar{L}(p^{1:T})$ using only the norm of local updates, and without imposing assumptions on the loss functions or how the local and global updates are made. As a result, the algorithm can be applied to design sampling distributions for essentially any stochastic optimization procedure. In Section 3, we show how our sampling method improves the upper bound of mini-batch SGD of non-convex objectives.

Suppose that at the beginning of the $t$-th communication round we know all $\{a_m^t\}_{m=1}^M$. Then the optimal sampling distribution

$$p_\star^t = (p_{\star,1}^t, \ldots, p_{\star,M}^t)^\top = \arg\min_{p \in \mathcal{P}_{M-1}} l_t(p)$$

is obtained as $p_{\star,m}^t = \sqrt{a_m^t}/(\sum_{m=1}^M \sqrt{a_m^t})$. Computing the distribution $p_\star^t$ is impractical as it requires local updates of all clients, which eradicates the need for client sampling. From the form of $p_\star^t$, we observe that clients with a large $a_m^t$ are more "important" and should have a higher probability of being selected. Since we do not know $\{a_m^t\}_{m=1}^M$, we will need to explore the environment to learn about the importance of clients before we can exploit the best strategy. Finally, we note that the relative importance of clients will change over time, which makes the environment dynamic and challenging.

Based on the above discussion, we cast the problem of creating a sequence of sampling distributions as an online learning task with bandit feedback, where a game is played between the server and environment. Let $p^1$ be the initial sampling distribution. At the beginning of iteration $t$, the server samples with replacement $K$ clients from $[M]$, denoted $S^t$, using $p^t$. The environment reveals $\{a_m^t\}_{m \in S^t}$ to the server, where $a_m^t = \lambda_m^2 \|g_m^t\|^2$. The environment also computes $l_t(p^t)$; however, this loss is not revealed to the server. The server then updates $p^{t+1}$ based on the feedback $\{\{a_m^u\}_{m \in S^u}\}_{u=1}^t$ and sampling distributions $\{p^u\}_{u=1}^t$. Note that in this game, the server only gets information about the chosen clients and, based on this partial information, or bandit feedback, needs to update the sampling distribution. On the other hand, we would like to be competitive with an oracle that can calculate the cumulative variance reduction loss. We will design $p^t$ in a way that is agnostic to the generation mechanism of $\{a^t\}_{t \geq 1}$, and will treat the environment as deterministic, with randomness coming only from $\{S^t\}_{t \geq 1}$. We describe an OSMD-based approach to solve this online learning problem.

## 2.2 OSMD Sampler

The variance-reduction loss function $l_t$ is a convex function on $\mathcal{P}_{M-1}$ and

$$\nabla l_t(q) = -\frac{1}{K}\left(\frac{a_1^t}{(q_1)^2}, \ldots, \frac{a_M^t}{(q_M)^2}\right)^\top \in \mathbb{R}^M \quad \text{for all} \quad q = (q_1, \ldots, q_M)^\top \in \mathbb{R}_{++}^M.$$

Since we do not observe $a^t$, we cannot compute $l_t(\cdot)$ or $\nabla l_t(\cdot)$. Instead, we can construct unbiased estimates of them. For any $q \in \mathcal{P}_{M-1}$, let $\hat{l}_t(q; p^t)$ be an estimate of $l_t(q)$ defined as

$$\hat{l}_t(q; p^t) = \frac{1}{K^2} \sum_{m=1}^M \frac{a_m^t}{q_m p_m^t} \mathcal{N}\left\{m \in S^t\right\}, \tag{6}$$

and $\nabla \hat{l}_t(q; p^t) \in \mathbb{R}^M$ has the $m$-th entry defined as

$$\left[\nabla \hat{l}_t(q; p^t)\right]_m = -\frac{1}{K^2} \cdot \frac{a_m^t}{q_m^2 p_m^t} \mathcal{N}\left\{m \in S^t\right\}. \tag{7}$$

The set $S^t$ is sampled with replacement from $[M]$ using $p^t$ and $\mathcal{N}\{m \in S^t\}$ denote the number of times that a client $m$ is chosen in $S^t$. Thus, $0 \leq \mathcal{N}\{m \in S^t\} \leq K$. Given $q$ and $p^t$, $\hat{l}_t(q; p^t)$ and $\nabla \hat{l}_t(q; p^t)$ are random variables in $\mathbb{R}$ and $\mathbb{R}^M$ that satisfy

$$\mathbb{E}_{S^t}\left[\hat{l}_t(q; p^t) \mid p^t\right] = l_t(q), \qquad \mathbb{E}_{S^t}\left[\nabla \hat{l}_t(q; p^t) \mid p^t\right] = \nabla l_t(q).$$

---

**Algorithm 1** OSMD Sampler

---

1: **Input:** A sequence of learning rates $\{\eta_t\}_{t\geq 1}$; parameter $\alpha \in (0, 1]$, $\mathcal{A} = \mathcal{P}_{M-1} \cap [\alpha/M, \infty)^M$; number of iterations $T$.
2: **Output:** $\hat{p}^{1:T}$.
3: **Initialize:** $\hat{p}^1 = p^{\text{unif}}$.
4: **for** $t = 1, 2, \ldots, T - 1$ **do**
5:     Sample $S^t$ by $\hat{p}^t$.
6:     Compute $\nabla \hat{l}_t(\hat{p}^t; \hat{p}^t)$ via equation 7.
7:     $\hat{p}^{t+1} = \arg\min_{p \in \mathcal{A}} \eta_t \langle p, \nabla \hat{l}_t(\hat{p}^t; \hat{p}^t) \rangle + D_\Phi(p \,\|\, \hat{p}^t)$.
8: **end for**

---

When $S^t$ and $p^t \in \mathbb{R}^M_{++}$ are given, $\hat{l}_t(q; p^t)$ is a convex function with respect to $q$ on $\mathbb{R}^M_{++}$ and satisfies $\hat{l}_t(q; p^t) - \hat{l}_t(q'; p^t) \leq \langle \nabla \hat{l}_t(q; p^t), q - q' \rangle$, for $q, q' \in \mathbb{R}^M_{++}$. The constructed estimates $\hat{l}_t(q; p^t)$ and $\nabla \hat{l}_t(q; p^t)$ are crucial for designing updates to the sampling distribution. To the best of our knowledge, while similar estimators as $\hat{l}_t(q; p^t)$ in equation 6 was used in previous literature (Borsos et al., 2018), we are the first one to propose $\nabla \hat{l}_t(q; p^t)$ in equation 7 and use it for updating sampling distribution.

OSMD Sampler is an online stochastic mirror descent algorithm for updating the sampling distribution, detailed in Algorithm 1. The sampling distribution is restricted to lie in the space $\mathcal{A} = \mathcal{P}_{M-1} \cap [\alpha/M, \infty)^M$, $\alpha \in (0, 1]$, to prevent the server from assigning small probabilities to devices. Let $\Phi : \mathcal{D} \subseteq \mathbb{R}^M \mapsto \mathbb{R}$ be a continuously differentiable convex function defined on $\mathcal{D}$, with $\mathcal{A} \subseteq \bar{\mathcal{D}}$. The learning rates $\{\eta_t\}_{t\geq 1}$ are positive and nonincreasing.[6] Line 7 of Algorithm 1 provides an update to the sampling distribution using the mirror descent update. The available feedback is used to construct an estimate of the loss, while the Bregman divergence between the current and next sampling distribution is used as a regularizer, ensuring that the updated sampling distribution does not change too much. The update only uses the most recent information, while forgetting the history, which results in nonstationarity of the sequence of sampling distributions. In Line 5 of Algorithm 1, we choose $S^t$ by sampling with replacement. In Section C.5, we discuss how to extend the results to sampling without replacement.

The mirror descent update in Line 7 is not available in a closed form in general and an iterative solver may be needed. However, when $\Phi(\cdot)$ is chosen as the negative entropy $\Phi_e(\cdot)$ (we use this as our default choice), a closed-form efficient solution can be obtained. An efficient implementation is shown in Algorithm 2 in Appendix C.1. The main cost comes from sorting the sequence $\{\tilde{p}^{t+1}_m\}^M_{m=1}$, which can be done with the computational complexity of $O(M \log M)$. However, note that we only update a few entries of $p^t$ to get $\tilde{p}^{t+1}$ and $p^t$ is sorted. Therefore, most entries of $\tilde{p}^{t+1}$ are also sorted. Using this observation, we can usually achieve a much faster running time, for example, by using an adaptive sorting algorithm (Estivill-Castro & Wood, 1992). Next, we provide a bound on the dynamic regret for OSMD Sampler.

## 2.3 DYNAMIC REGRET OF OSMD SAMPLER

We first describe the dynamic regret used to measure the performance of an online algorithm that generates a sequence of sampling distributions $\{\hat{p}\}_{t\geq 1}$ in a non-stationary environment. Given any comparator sequence $q^{1:T} \in \mathcal{P}^T_{M-1}$, the dynamic regret is defined as

$$\text{D-Regret}_T(q^{1:T}) = \bar{L}\left(\hat{p}^{1:T}\right) - \bar{L}\left(q^{1:T}\right). \tag{8}$$

In contrast, the static regret measures the performance of an algorithm relative to the best fixed sampling distribution, that is, it restricts $q^1 = \cdots = q^T$ (Namkoong et al., 2017; Salehi et al., 2017; Borsos et al., 2018; 2019). When using a fixed comparator $q^1 = \cdots = q^T = q$, we write the regret as $\text{D-Regret}_T(q)$; besides, we write $\text{D-Regret}_T$ to denote $\text{D-Regret}_T(p^{1:T}_\star)$.

---

[6]We use the term *learning rate* when discussing an online algorithm that learns a sampling distribution, while the term *stepsize* is used in the context of an optimization algorithm.

The following quantity describes the dynamic complexity of a comparator sequence and will appear in the regret bound below.

**Definition 2.1** (Total Variation). The total variation of a comparator sequence $q^{1:T}$ with respect to the norm $\|\cdot\|$ on $\mathbb{R}^M$ is $\mathrm{TV}\left(q^{1:T}\right) = \sum_{t=1}^{T-1} \|q^{t+1} - q^t\|$.

The total variation measures how variable a sequence is. The larger the total variation $\mathrm{TV}(q^{1:T})$, the more variable $q^{1:T}$ is, and such a comparator sequence is harder to match. To give an upper bound on the dynamic regret of OSMD Sampler, we need the following assumptions.

**Assumption 1.** *The function $\Phi(\cdot)$ is $\rho$-strongly convex, $\rho > 0$ with respect to $\|\cdot\|$:*

$$\Phi(x) \geq \Phi(y) + \langle \nabla\Phi(y), x - y \rangle + \frac{\rho}{2}\|x - y\|^2, \quad x, y \in \mathcal{D}.$$

**Assumption 2.** *There exist positive functions $H(M, \alpha)$, $\{Q_t(M, \alpha)\}_{t \geq 1}$, and $D_{\max}(M, \alpha)$, such that $\|\nabla\Phi(p)\|_\star \leq H(M, \alpha)$, $\|\nabla\hat{l}_t(p; q)\|_\star \leq Q_t(M, \alpha)$ almost surely, and $D_\Phi(q \,\|\, p^{\mathrm{unif}}) \leq D_{\max}(M, \alpha)$ for all $p, q \in \mathcal{A}$.*

These assumptions are standard in the literature (Hall & Willett, 2015). When we choose $\Phi = \Phi_e$ and $\|\cdot\| = \|\cdot\|_1$, these assumptions hold (See Appendix D.3). Under Assumption 1, we have $D_\Phi(x \,\|\, y) \geq (\rho^2/2)\|x - y\|^2$. To simplify the notation, we will omit $M$ and $\alpha$ from $H$, $\{Q_t\}_{t \geq 1}$, and $D_{\max}$. We also need the following quantities that quantify how far $q^t$ is from $\mathcal{A}$. Given $q^t \in \mathcal{P}_{M-1}$ and $\alpha \in (0, 1]$, let

$$\psi(q^t, \alpha) := \sum_{m=1}^{M}\left(\frac{\alpha}{M} - q_m^t\right)\mathbb{1}\left\{q_m^t < \frac{\alpha}{M}\right\}, \qquad \omega(q^t, \alpha) := \frac{\sum_{m=1}^{M}\left(\frac{\alpha}{M} - q_m^t\right)\mathbb{1}\left\{q_m^t < \frac{\alpha}{M}\right\}}{\sum_{m=1}^{M}\left(q_m^t - \frac{\alpha}{M}\right)\mathbb{1}\left\{q_m^t \geq \frac{\alpha}{M}\right\}},$$

$$\phi(q^t, \alpha) := \frac{\omega(q^t, \alpha)}{1 - \omega(q^t, \alpha)\left(1 - \frac{\alpha}{M}\right)}. \tag{9}$$

We will use these quantities to characterize the projection error in the following theorem, which is the main result of this section.

**Theorem 1.** *Suppose Assumptions 1-2 hold and we use $\|\cdot\| = \|\cdot\|_1$ to define the total variation. Assume that $\{\eta_t\}_{t \geq 1}$ is a nonincreasing sequence. Let $\hat{p}^{1:T}$ be a sequence generated by Algorithm 1. For any comparator sequence $q^{1:T}$, where $q^t$ is allowed to be deterministic or random, we have*

$$\mathit{D\text{-}Regret}_T(q^{1:T}) \leq \underbrace{\frac{D_{\max}}{\eta_1} + \frac{2H}{\eta_T}\mathbb{E}\left[\mathit{TV}\left(q^{1:T}\right)\right] + \frac{2}{\rho}\sum_{t=1}^{T}\eta_t\mathbb{E}\left[Q_t^2\right]}_{\text{Intrinsic Regret}} +$$

$$\underbrace{\frac{8H}{\eta_T}\sum_{t=1}^{T}\mathbb{E}\left[\psi(q^t, \alpha)\right] + \sum_{t=1}^{T}\mathbb{E}\left[\phi(q^t, \alpha)l_t(q^t)\right]}_{\text{Projection Error}}.$$

*Proof.* The major challenge of the proof is to construct a projection of the comparator sequence $q^{1:T}$ onto $\mathcal{A}^T$ and bound the projection error. To the best of our knowledge, this bound on the projection error of a dynamic sequence is novel. Another challenge is to deal with the dynamic comparator, which requires us to connect the cumulative regret with the total variation of the comparator sequence. See Appendix D.2 for more details. $\qquad\square$

From Theorem 1, we see that the bound on the dynamic regret consists of two parts. The first part is the intrinsic regret, quantifying the difficulty of tracking a comparator sequence in $\mathcal{A}^T$; the second part is the projection error, arising from projecting the comparator sequence onto $\mathcal{A}^T$. Note that the intrinsic regret depends on $\alpha$ through $D_{\max}$, $H$, and $\{Q_t\}_{t \geq 1}$. As shown in Appendix D.2, we have $0 \leq \omega(q^t, \alpha) \leq 1$ for all $\alpha \in [0, 1]$, which implies that $\phi(q^t, \alpha) \leq M/\alpha$. Furthermore, $\psi(q^t, \alpha) \leq \sum_{m=1}^{M}(\alpha/M)\mathbb{1}\{q_m^t < (\alpha/M)\} \leq \alpha$. Therefore, the projection error can be upper bounded by $(8H\alpha)/\eta_T + (M/\alpha)\sum_{t=1}^{T}l_t(q^t)$. More importantly, when $q_m^t \in \mathcal{A}$, we have $\psi(q^t, \alpha) = \omega(q^t, \alpha) =$

$\phi(q^t, \alpha) = 0$. Thus, when the comparator sequence belongs to $\mathcal{A}^T$, the projection error vanishes and we only have the intrinsic regret. As $\alpha$ decreases from one to zero, the intrinsic regret gets larger (it often tends to infinity as shown in Corollary 3), while we are allowing a larger class of comparator sequences; on the other hand, the projection error decreases to zero, since the gap between $\mathcal{A}$ and $\mathcal{P}_{M-1}$ vanishes with $\alpha$. An optimal choice of $\alpha$ balances the two sources of regret.

When $\|\cdot\| = \|\cdot\|_1$ and $\Phi$ is the unnormalized negative entropy, the Step 7 of Algorithm 1 has a closed-form solution (see Proposition 1). By Pinsker's inequality, the unnormalized negative entropy $\Phi_e(x)$ is 1-strongly convex on $\mathcal{P}_{M-1}$ with respect to $\|\cdot\|_1$. When $q^{1:T}$ is a deterministic sequence, we have $\mathbb{E}\left[\text{TV}\left(q^{1:T}\right)\right] = \text{TV}\left(q^{1:T}\right)$.

With Theorem 1, we have the following corollary.

**Corollary 1.** *Suppose conditions of Theorem 1 hold and let $\hat{p}^{1:T}$ be the sequence generated by Algorithm 1. For any comparator sequence $q^{1:T}$, we choose $\alpha$ such that $q^t \in \mathcal{A}$ for all $t \in [T]$. Let*

$$\eta = \frac{K^2 \alpha^3}{M^3} \sqrt{\frac{\log M + 2\log(M/\alpha)\mathbb{E}\left[\text{TV}(q^{1:T})\right]}{2\sum_{t=1}^{T} \mathbb{E}\left[(\bar{a}^t)^2\right]}}. \tag{10}$$

*Then*

$$\text{D-Regret}_T(q^{1:T}) \leq \frac{2\sqrt{2}M^3}{K^2\alpha^3} \sqrt{\left[\log M + 2\log\left(M/\alpha\right)\mathbb{E}\left[\text{TV}(q^{1:T})\right]\right] \sum_{t=1}^{T} \mathbb{E}\left[\left(\bar{a}^t\right)^2\right]},$$

*where $\bar{a}^t \coloneqq \max_{1 \leq m \leq M} \lambda_m^2 \|g_m^t\|^2 = \max_{1 \leq m \leq M} a_m^t$ for all $t \in [T]$.*

*Proof.* The proof follows directly from Corollary 3 in Appendix D.3. □

Note that as the training proceeds, the norms of local updates are decreasing, thus $\{\bar{a}^t\}_{t=1}^T$ is typically a decreasing sequence. Thus, a naive upper bound is $\sum_{t=1}^{T} \mathbb{E}[(\bar{a}^t)^2] = O(T)$. However, in Appendix F.6, we empirically show that $\bar{a}^t$ decreases fast and the cumulative square sum of this sequence will converge to a constant, that is, $\sum_{t=1}^{T} \mathbb{E}[(\bar{a}^t)^2] = O(1)$. This result further implies that the static regret with respect to the best fixed sampling distribution in hindsight (where $TV(q^{1:T}) = 0$) is empirically $O(1)$, which is much better than the rates in previous research (Salehi et al., 2017; Borsos et al., 2018). Besides, our regret analysis also allows for sampling distributions that change over time (where $TV(q^{1:T}) > 0$), and thus is more general than previous results.

The choice of learning rate in equation 10 depends on unknown quantities prior to training, and is thus impractical. In Appendix C.3 and Appendix C.4, we introduce practical automatic parameter tuning strategies with the help of online ensemble method and doubling trick.

## 3 CONVERGENCE ANALYSIS OF MINI-BATCH SGD WITH OSMD SAMPLER

We illustrate how OSMD Sampler can be used to provably improve the convergence rate of the minibatch SGD. The detailed algorithm is given in Algorithm 3 in Appendix C.2. We use mini-batch SGD as a motivating example to show how adaptive sampling improves the convergence guarantee of an optimization algorithm. The analysis in this section can be extended to other optimization algorithms as well.

To simplify the notation, we denote $F_m(w) \coloneqq \phi(w; \mathcal{D}_m)$ and let $\lambda_m = 1/M$ for all $m \in [M]$. We assume that $w \in \mathcal{W} \subset \mathbb{R}^d$, where $\mathcal{W}$ is a compact set. Besides, we assume that client objectives are differentiable and $L$-smooth functions.

**Assumption 3.** *For all $m \in [M]$, $F_m(\cdot)$ is differentiable and $L$-smooth, that is,*

$$\|\nabla F_m(x) - \nabla F_m(y)\| \leq L\|x - y\|, \quad \text{for all } x, y \in \mathcal{W}.$$

Note that we allow $F_m(\cdot)$ to be non-convex. We also assume that the objective function $F(\cdot)$ is lower-bounded, that is, we assume that $F^\star \coloneqq \inf_{w \in \mathcal{W}} F(w) > -\infty$. In addition, we make the following assumption about the local stochastic gradient.

**Assumption 4.** *For all $w \in \mathcal{W}$ and $m \in [M]$, we have $\mathbb{E}_{\xi \sim \mathcal{D}_m} [\nabla \phi(w; \xi)] = \nabla F_m(w)$ and $\mathbb{E}_{\xi \sim \mathcal{D}_m} \left[ \|\nabla \phi(w; \xi) - \nabla F_m(w)\|^2 \right] \leq \sigma^2$.*

Next, we introduce quantities that characterize heterogeneity of the optimization problem. Specifically, heterogeneity characterizes how objective functions of different clients differ from each other. In a federated learning problem, heterogeneity can be large and it is important to understand its effect on the convergence of algorithms. The following three quantities characterize the heterogeneity:

$$\zeta_0^2 := \sup_{w \in \mathcal{W}} \frac{1}{M} \sum_{m=1}^{M} \|\nabla F_m(w) - \nabla F(w)\|^2 = \sup_{w \in \mathcal{W}} \left\{ \frac{1}{M} \sum_{m=1}^{M} \|\nabla F_m(w)\|^2 - \|\nabla F(w)\|^2 \right\},$$

$$\zeta_1^2 := \min_{p \in \mathcal{P}_{M-1}} \sup_{w \in \mathcal{W}} \left\{ \frac{1}{M^2} \sum_{m=1}^{M} \frac{1}{p_m} \|\nabla F_m(w)\|^2 - \|\nabla F(w)\|^2 \right\},$$

$$\zeta_2^2 := \sup_{w \in \mathcal{W}} \left\{ \left( \frac{1}{M} \sum_{m=1}^{M} \|\nabla F_m(w)\| \right)^2 - \|\nabla F(w)\|^2 \right\}.$$

By Jensen's inequality we have that $\zeta_0 \geq \zeta_1 \geq \zeta_2$ and we assume that $\zeta_0 < \infty$. The quantity $\zeta_0$ has been commonly used to quantify first-order heterogeneity (Karimireddy et al., 2020a;b), while $\zeta_1$ and $\zeta_2$ are variants of $\zeta_0$ corresponding to different sampling schemes. More specifically, when we use $q^\star$ and $p_\star^{1:T}$ to sample clients, we will have the heterogeneity level to be $\zeta_1$ and $\zeta_2$ respectively, where $q^\star = \arg \min_{p \in \mathcal{P}_{M-1}} \sup_{w \in \mathcal{W}} \sum_{m=1}^{M} (1/p_m) \|\nabla F_m(w)\|^2$ is the best fixed sampling distribution. Finally, the following quantities are useful in stating the convergence guarantee. Recall that $B$ is the local batch size in Algorithm 3 and $K = |S^t|$. Let $D^F := F(w^0) - F^\star$,

$$R_0 := \frac{D^F L}{T} + \frac{\sigma \sqrt{D^F L}}{\sqrt{TKB}} + \frac{\zeta_0 \sqrt{D^F L}}{\sqrt{TK}},$$

$$R_1 := \frac{D^F L}{T} + \frac{\sigma \sqrt{D^F L}}{\sqrt{TKB\alpha}} + \frac{\zeta_1 \sqrt{D^F L}}{\sqrt{TK}} + \frac{\sqrt{D^F L} \sqrt{\text{D-Regret}_T(q^\star))}}{T},$$

$$R_2 := \frac{D^F L}{T} + \frac{\sigma \sqrt{D^F L}}{\sqrt{TKB\alpha}} + \frac{\zeta_2 \sqrt{D^F L}}{\sqrt{TK}} + \frac{\sqrt{D^F L} \sqrt{\text{D-Regret}_T(p_\star^{1:T}))}}{T}.$$

We are now ready to give the convergence guarantee of Algorithm 3.

**Theorem 2.** *Assume Assumption 3 and Assumption 4 hold. Let $\mu_t = \mu$, $t \in [T]$, where $\mu$ is given in equation 33 in Appendix. Let $\{w^0, \dots, w^{T-1}\}$ be the sequence of iterates generated by Algorithm 3 and let $w^R$ denote an element of that sequence chosen uniformly at random. When $q^\star \in \mathcal{A}$, we have $\mathbb{E} \left[ \|\nabla F(w^R)\|^2 \right] \lesssim R_1$; when $p_\star^t \in \mathcal{A}$ for all $t \in [T]$, we have $\mathbb{E} \left[ \|\nabla F(w^R)\|^2 \right] \lesssim R_2$; when $q^\star \in \mathcal{A}$ and $p_\star^t \in \mathcal{A}$ both hold, we have $\mathbb{E} \left[ \|\nabla F(w^R)\|^2 \right] \lesssim \min\{R_1, R_2\}.$*

See proof in Appendix D.6. We derive different convergence rates $R_1$ and $R_2$ in Theorem 2 by choosing different comparators. More specifically, $R_1$ is derived by comparing against $q^\star$ and $R_2$ is derived by comparing against $p_\star^{1:T}$. *The different notions of heterogeneity reveal the fact that different sampling schemes can change the convergence speed of optimization algorithms through the change of heterogeneity level.*

Note that $R_0$ is the rate of mini-batch SGD under uniform sampling (Ghadimi & Lan, 2013). OSMD Sampler can obtain tighter rates than uniform sampling when $R_1$ or $R_2$ are smaller than $R_0$. To have $R_1 \lesssim R_0$, we need $(\sigma/\sqrt{KB})(1/\sqrt{\alpha} - 1) + \sqrt{\text{D-Regret}_T(q^\star)/T} \lesssim (1/\sqrt{K})(\zeta_0 - \zeta_1)$. By Corollary 1, we have a worst-case upper bound of D-Regret$_T(q^\star)$ as $O(\sqrt{T})$; empirical evidence in Appendix F.6 suggests tighter rates $O(1)$. With either rates, we always have D-Regret$_T(q^\star)/T = o(1)$. Thus, to have $R_1 \lesssim R_0$, we only need $(\sigma/\sqrt{B})(1/\sqrt{\alpha} - 1) \ll \zeta_0 - \zeta_1$ asymptotically. That is, we want the gap between the heterogeneity under best fixed sampling distribution and uniform sampling to be large, compared to $(\sigma/\sqrt{B})(1/\sqrt{\alpha} - 1)$, which is always true when we use full local gradient, i.e., when $\sigma = 0$. Similar arguments apply to when $R_2 \lesssim R_0$. See a more detailed discussion in Appendix A.2.

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

# A   MORE DISCUSSIONS

In this section, we include additional discussions that is omitted from the main text due to space limit.

## A.1   DETAILED DISCUSSION ABOUT CONTRIBUTIONS

We develop an algorithm based on OSMD that generates a sequence of sampling distributions $\{p^t\}_{t\geq 1}$ based on the partial feedback available to the server from the sampled clients. We prove a bound on regret relative to the any *dynamic* comparators, which allows us to consider the best sequence of sampling distributions as they change over iterations. The bound includes a *total variation* term that characterizes the *intrinsic difficulty* of the problem by capturing the difficulty of following the best sequence of distributions. Such a characterization of problem difficulty is novel. Besides, our theoretical result can recover the results in previous research as special cases and is thus strictly more general. Moreover, we theoretically improve the convergence guarantee of optimization algorithm by using our sampling scheme over uniform sampling. We show that adaptive sampling can help reduce the dependency on the heterogeneity level of the problem.

We demonstrate the empirical superiority of the proposed algorithm through synthetic and real data experiments. In addition to client sampling in FL, our proposed algorithm will have a broad impact on stochastic optimization. Adapting our algorithm to any stochastic optimization procedure that chooses samples, such as SGD, or coordinates, such as Stochastic Coordinate Descent, may improve their performance. For example, Zhao et al. (2022) illustrated the practical benefits of our algorithm to speed up L-SVRG and L-Katyusha.

The learning rate of the proposed algorithm depends on the total variation of the comparator sequence, which is generally unknown. Furthermore, it also depends on the total number of iterations that can also be unknown. In appendix, we make the algorithm practical by addressing a few technical challenges. In particular, we adapt the follow-the-regularized-leader algorithm and doubling trick to *automatically* choose the learning rate that performs as well as the best learning rate asymptotically.

## A.2   DETAILED DISCUSSION ABOUT THEOREM 2

We derive different convergence rates $R_1$ and $R_2$ in Theorem 2 by choosing different comparators, which rates can be applied shall depend on the tuning parameter $\alpha$ (or equivalently the space $\mathcal{A}$). More specifically, $R_1$ is derived by comparing against $q^\star$ and $R_2$ is derived by comparing against $p_\star^{1:T}$. *The different notions of heterogeneity reveals the fact that different sampling schemes can change the convergence speed of optimization algorithms through the change of heterogeneity level.*

It is worth noting that one can always choose other comparator sequences besides these two and derived new upper bounds. When choosing a more flexible comparator, one can obtain a smaller heterogeneity, but risk requiring higher regret and larger space $\mathcal{A}$ (which will require smaller $\alpha$). A good choice of comparator should keep a trade-off between these two concerns. We leave how to choose the optimal comparator sequence as a future research direction.

Note that $R_0$ is the rate of mini-batch SGD under uniform sampling (Ghadimi & Lan, 2013). To see when OSMD Sampler can obtain tighter rates than uniform sampling, we only need $R_1$ or $R_2$ to be smaller than $R_0$.

**When will $R_1 \ll R_0$.** To have $R_1 \lesssim R_0$, we need $(\sigma/\sqrt{KB})(1/\sqrt{\alpha}-1)+\sqrt{\text{D-Regret}_T(q^\star)/T} \lesssim (1/\sqrt{K})(\zeta_0 - \zeta_1)$. By Corollary 1, we have a worst-case upper bound of D-Regret$_T(q^\star)$ as $O(\sqrt{T})$; empirical evidence in Appendix F.6 suggests tighter rates $O(1)$. With either rates, we always have D-Regret$_T(q^\star)/T = o(1)$. Thus, to have $R_1 \lesssim R_0$, we only need $(\sigma/\sqrt{B})(1/\sqrt{\alpha}-1) \ll \zeta_0 - \zeta_1$ asymptotically. That is, we want the gap between the heterogeneity under best fixed sampling distribution and uniform sampling be large, compared to $(\sigma/\sqrt{B})(1/\sqrt{\alpha}-1)$. When $\sigma = 0$, that is, if we use full local gradient, then this is always true.

**When will $R_2 \ll R_0$.** Similarly, to have $R_2 \ll R_0$, we need $(\sigma/\sqrt{KB})(1/\sqrt{\alpha}-1) + \sqrt{\text{D-Regret}_T(p_\star^\star)/T} \lesssim (1/\sqrt{K})(\zeta_0 - \zeta_2)$.. A worst-case upper bound of D-Regret$_T(p_\star^{1:T})$

based on Corollary 1 is $O(T)$; however, as shown in Appendix F.6, the empirical evidence suggests that we may have D-Regret$_T(p_\star^{1:T})$ to be $O(\sqrt{T})$ or even $O(1)$, which then implies that D-Regret$_T(p_\star^{1:T})/T = o(1)$. Given this is true, to have $R_1 \lesssim R_0$, we need $(\sigma/\sqrt{B})(1/\sqrt{\alpha} - 1) \ll \zeta_0 - \zeta_1$ asymptotically. That is, we want the gap between the heterogeneity under best dynamic sampling distribution and uniform sampling be large, compared to $(\sigma/\sqrt{B})(1/\sqrt{\alpha} - 1)$. When $\sigma = 0$, that is, if we use full local gradient, then this is always true.

**New characterization of heterogeneity.** Let $\Delta\zeta_i = \zeta_0 - \zeta_i$ for $i = 1, 2$. Based on the previous analysis, we see that $\Delta\zeta_i$, $i = 1, 2$, can also help measure the dissimilarity of the different clients. Compared with $\zeta_0$, the new characterization reflects how much gain we can have by using non-uniform sampling, where $\Delta\zeta_1$ measures the advantage gained by fixed sampling distribution, and $\Delta\zeta_2$ measures the advantage gained by dynamic sampling distribution.

# B   MORE RELATED WORK

Our paper is also closely related to importance sampling in stochastic optimization. Zhao & Zhang (2015); Needell et al. (2016) illustrated that by sampling observations from a nonuniform distribution when using a gradient-based stochastic optimization method, one can achieve faster convergence. They designed a fixed sampling distribution using prior knowledge on the upper bounds of gradient norms. Csiba & Richtárik (2018) extended the importance sampling to mini-batches. Stich et al. (2017); Johnson & Guestrin (2018); Gopal (2016) developed adaptive sampling strategies that allow the sampling distribution to change over time. Nesterov (2012); Perekrestenko et al. (2017); Zhu et al. (2016); Salehi et al. (2018) discussed importance sampling in stochastic coordinate descent methods. Namkoong et al. (2017); Salehi et al. (2017); Borsos et al. (2018; 2019); Hanchi & Stephens (2020) illustrated how to design the sampling distribution by solving an online learning task with bandit feedback. Namkoong et al. (2017); Salehi et al. (2017) designed the sampling distribution by solving a multi-armed bandit problem with the EXP3 algorithm (Lattimore & Szepesvári, 2020, Chapter 11). Borsos et al. (2018) used the follow-the-regularized-leader algorithm (Lattimore & Szepesvári, 2020, Chapter 28) to solve an online convex optimization problem and make updates to the sampling distribution. Borsos et al. (2019) restricted the sampling distribution to be a linear combination of distributions in a predefined set and used an online Newton step to make updates to the mixture weights. The above approaches estimate a stationary distribution, while the best distribution is changing with iterations and, therefore, is intrinsically dynamic. In addition to having suboptimal empirical performance, these papers provide theoretical results that only establish a regret relative to a fixed sampling distribution in hindsight. To address this problem, Hanchi & Stephens (2020) took a non-stationary approach where the most recent information for each client was kept. A decreasing stepsize sequence is required to establish a regret bound. Furthermore, the regret bound does not capture the intrinsic difficulty of the problem. In comparison, we establish a regret bound relative to a dynamic comparator—a sequence of sampling distributions—without imposing assumptions on the stepsize sequence, and this bound includes the dependence on the total variation term characterizing the intrinsic difficulty of the problem.

Our paper also contributes to the literature on online convex optimization. We cast the client sampling problem as an online learning problem (Hazan, 2016) and adapt algorithms from the dynamic online convex optimization literature to solve it. Hall & Willett (2015); Yang et al. (2016); Daniely et al. (2015) proposed methods that achieve sublinear dynamic regret relative to dynamic comparator sequences. In particular, Hall & Willett (2015) used a dynamic mirror descent algorithm to achieve sublinear dynamic regret with total variation characterizing the intrinsic difficulty of the environment. However, the optimal tuning parameters depend on the unknown total variation. On the other hand, van Erven & Koolen (2016); Zhang et al. (2018) proposed different online ensemble approaches to automatically choose the tuning parameters for online gradient descent. Compared with the problem settings in the above studies, there are two key new challenges that we need to address. First, we only have partial information—bandit feedback—instead of the full information about the loss functions. Second, the loss functions in our case are unbounded, which violates the common boundedness assumption in the online learning literature. To overcome the first difficulty, we construct an unbiased estimator of the loss function and its gradient, which are then used to make an update to the sampling distribution. We address the second challenge by first bounding the regret of our algorithm when the sampling distributions in the comparator sequence lie in a region of the

simplex for which the loss is bounded, and subsequently analyze the additional regret introduced by projecting the elements of the comparator sequence to this region.

## C ADDITIONAL ALGORITHMS

### C.1 ALGORITHM TO SOLVE STEP 7 OF ALGORITHM 1 WHEN $\Phi$ IS UNNORMALIZED NEGATIVE ENTROPY

When $\Phi(\cdot)$ is chosen as the negative entropy $\Phi_e(\cdot)$, a closed-form efficient solution can be obtained as shown in Proposition 1.

**Proposition 1.** *Suppose $\Phi = \Phi_e$ is the unnormalized negative entropy in Algorithm 1. Let*

$$\tilde{p}_m^{t+1} = p_m^t \exp\left\{\mathcal{N}\left\{m \in S^t\right\} \eta_t a_m^t / (K^2 (p_m^t)^3)\right\}, \qquad m \in [M].$$

*Let $\pi : [M] \mapsto [M]$ be a permutation such that $\tilde{p}_{\pi(1)}^{t+1} \leq \tilde{p}_{\pi(2)}^{t+1} \leq \cdots \leq \tilde{p}_{\pi(M)}^{t+1}$. Let $m_\star^t$ be the smallest integer $m$ such that*

$$\tilde{p}_{\pi(m)}^{t+1}\left(1 - \frac{m-1}{M}\alpha\right) > \frac{\alpha}{M}\sum_{j=m}^{M}\tilde{p}_{\pi(j)}^{t+1}.$$

*Then*

$$\hat{p}_m^{t+1} = \begin{cases} \alpha/M & \text{if } \pi(m) < m_\star^t \\ \left((1 - ((m_\star^t - 1)/M)\alpha)\tilde{p}_m^{t+1}\right) / \left(\sum_{j=m_\star^t}^{M}\tilde{p}_{\pi(j)}^{t+1}\right) & \text{otherwise.} \end{cases}$$

*Proof.* See Appendix D.1. □

An efficient implementation is shown in Algorithm 2.

---

**Algorithm 2** Solver of Step 7 of Algorithm 1 when $\Phi$ is unnormalized negative entropy

1: **Input:** $\hat{p}^t$, $S^t$, $\{a_m^t\}_{m \in S^t}$, and $\mathcal{A} = \mathcal{P}_{M-1} \cap [\alpha/M, \infty)^M$.
2: **Output:** $\hat{p}^{t+1}$.
3: Let $\tilde{p}_m^{t+1} = p_m^t \exp\left\{\mathcal{N}\left\{m \in S^t\right\} \eta_t a_m^t / (K^2 (p_m^t)^3)\right\}$ for $m \in [M]$.
4: Sort $\{\tilde{p}_m^{t+1}\}_{m=1}^M$ in a non-decreasing order: $\tilde{p}_{\pi(1)}^{t+1} \leq \tilde{p}_{\pi(2)}^{t+1} \leq \cdots \leq \tilde{p}_{\pi(M)}^{t+1}$.
5: Let $v_m = \tilde{p}_{\pi(m)}^{t+1}\left(1 - \frac{m-1}{M}\alpha\right)$ for $m \in [M]$.
6: Let $u_m = \frac{\alpha}{M}\sum_{j=m}^{M}\tilde{p}_{\pi(j)}^{t+1}$ for $m \in [M]$.
7: Find the smallest $m$ such that $v_m > u_m$, denoted as $m_\star^t$.
8: Let $\hat{p}_m^{t+1} = \begin{cases} \alpha/M & \text{if } \pi(m) < m_\star^t \\ \left((1 - ((m_\star^t - 1)/M)\alpha)\tilde{p}_m^{t+1}\right) / \left(\sum_{j=m_\star^t}^{M}\tilde{p}_{\pi(j)}^{t+1}\right) & \text{otherwise.} \end{cases}$

---

### C.2 MIN-BATCH SGD WITH OSMD SAMPLER

In this section, we describe the Min-batch SGD with OSMD Sampler in Algorithm 3. Compared to classical min-batch SGD, the key ingredient of Algorithm 3 is Line 13, where the server updates the sampling distribution by OSMD Sampler, and Line 5, where the server samples the local mini-batch from a non-uniform sampling distribution.

### C.3 ADAPTIVE-OSMD SAMPLER

The choice of the sequence of learning rates $\{\eta_t\}_{t \geq 1}$ has a large effect on the performance of OSMD Sampler. Similar to Corollary 1, we can have the following Corollary.

---

**Algorithm 3** Min-batch SGD with OSMD Sampler

---

1: **Input:** Number of communication rounds $T$, number of clients chosen in each round $K$, local batch size $B$, initial model parameter $w^0$, stepsize $\{\mu_t\}_{t=0}^{T-1}$.
2: **Output:** The final model parameter $w^T$.
3: **Initialize:** $\hat{p}^0 = p^{\text{unif}}$.
4: **for** $t = 0, 1, \ldots, T-1$ **do**
5:   Sample $S^t$ with replacement from $[M]$ with probability $\hat{p}^t$, such that $|S^t| = K$.
6:   **for** $m \in S^t$ **do**
7:     Download the current model parameter $w^t$.
8:     Locally sample a mini-batch $B_m^t = \{\xi_{m,1}^t, \ldots, \xi_{m,B}^t\}$ i.i.d. uniformly random from $[n_m]$.
9:     Locally compute and upload $g_m^t = (1/B) \sum_{b=1}^{B} \nabla \phi(w^t; \xi_{m,b}^t)$ to the server.
10:   **end for**
11:   Server computes $a_m^t = \lambda_m^2 \|g_m^t\|^2$ for $m \in S^t$ and

$$g^t = \frac{1}{K} \sum_{m \in S^t} \frac{\lambda_m}{\hat{p}_m^t} g_m^t.$$

12:   Server makes update of the model parameter $w^{t+1} \leftarrow w^t - \mu_t g^t$.
13:   Server obtains updated sampling distribution $\hat{p}^{t+1}$ by Algorithm 1.
14: **end for**

---

**Corollary 2.** *Suppose conditions of Theorem 1 hold and let $\hat{p}^{1:T}$ be the sequence generated by Algorithm 1. We choose $\alpha$ such that $p_\star^t \in \mathcal{A}$ for all $t \in [T]$. Let*

$$\eta = \frac{K^2 \alpha^3}{M^3 \sqrt{\mathbb{E}\left[(\bar{a}^1)^2\right]}} \sqrt{\frac{\log M + 2\log(M/\alpha)\mathbb{E}\left[TV(p_\star^{1:T})\right]}{2T}}. \tag{11}$$

*Then*

$$\text{D-Regret}_T(p_\star^{1:T}) \leq \frac{2\sqrt{2}M^3 \sqrt{\mathbb{E}\left[(\bar{a}^1)^2\right]}}{K^2 \alpha^3} \sqrt{\left[\log M + 2\log(M/\alpha)\mathbb{E}\left[TV(p_\star^{1:T})\right]\right] T}, \tag{12}$$

*where $\bar{a}^t := \max_{1 \leq m \leq M} a_m^t = \max_{1 \leq m \leq M} \lambda_m^2 \|g_m^t\|^2$.*

The choice in equation 11 still depends on unknown quantities such as $\mathbb{E}[\bar{a}^1]$, $\mathbb{E}[\text{TV}(p_\star^{1:T})]$, and $T$. To get $\mathbb{E}[\bar{a}^1]$, we can add a pre-training phase where we broadcast the initial model parameter $w^0$ to all devices before the start of the training, and collect the returned $\|g_m^0\|^2$ from all responsive devices, which we denote as $S^0$. Then $\hat{\bar{a}}^1 := \max_{m \in S^0} \lambda_m \|g_m^0\|^2$. On the other hand, $\mathbb{E}[\text{TV}(p_\star^{1:T})]$ and $T$ are hard to estimate in advance of the training. We discuss how to use an online ensemble method to choose the learning rate without the knowledge of $\mathbb{E}[\text{TV}(p_\star^{1:T})]$, and how to get rid of the dependence on $T$ using the doubling trick.

The main idea is to run a set of expert algorithms, each with a different learning rate for Algorithm 1. We then use a prediction-with-expert-advice algorithm to track the best performing expert algorithm.[7] More specifically, we define the set of expert learning rates as

$$\mathcal{E} := \left\{ 2^{e-1} \cdot \frac{K^2 \alpha^3}{M^3 \sqrt{\mathbb{E}\left[(\bar{a}^1)^2\right]}} \sqrt{\frac{\log M}{2T}} \; \middle| \; e = 1, 2, \ldots, E \right\}, \tag{13}$$

where

$$E = \lfloor \frac{1}{2} \log_2 \left( 1 + \frac{4\log(M/\alpha)}{\log M}(T-1) \right) \rfloor + 1. \tag{14}$$

Then for each $\eta_e \in \mathcal{E}$, Adaptive-OSMD Sampler algorithm runs an expert algorithm to generate a sequence of sampling distributions $\hat{p}_e^{1:T}$. Meanwhile, it also runs a meta-algorithm that uses

---

[7]We refer the reader to Chapter 2 of Cesa-Bianchi & Lugosi (2006) for an overview of prediction-with-expert-advice algorithms.

---

**Algorithm 4** Adaptive-OSMD Sampler

---

1: **Input:** Meta learning rate $\gamma$; the set of expert learning rates $\mathcal{E} = \{\eta_1 \leq \eta_2 \leq \cdots \leq \eta_E\}$ with $E = |\mathcal{E}|$; parameter $\alpha \in (0,1]$, $\mathcal{A} = \mathcal{P}_{M-1} \cap [\alpha/M, \infty)^M$; number of iterations $T$; initial distribution $p^{\text{init}}$.
2: **Output:** $\hat{p}^{1:T}$.
3: Set $\theta_e^1 = (1 + 1/E)/(e(e+1))$ and $\hat{p}_e^1 = p^{\text{init}}$, $e \in [E]$.
4: **for** $t = 1, 2, \ldots, T-1$ **do**
5:     Compute $\hat{p}^t = \sum_{e=1}^E \theta_e^t \hat{p}_e^t$.
6:     Sample $S^t$ by $\hat{p}^t$.
7:     **for** $e = 1, 2, \ldots, E$ **do**
8:         Compute $\hat{l}_t(\hat{p}_e^t; \hat{p}^t)$ via equation 6 and $\nabla \hat{l}_t(\hat{p}_e^t; \hat{p}^t)$ via equation 7.
9:         Solve $\hat{p}_e^{t+1} = \arg\min_{p \in \mathcal{A}} \eta_e \langle p, \nabla \hat{l}_t(\hat{p}_e^t; \hat{p}^t) \rangle + D_\Phi(p \,\|\, \hat{p}_e^t)$ via Algorithm 2.
10:     **end for**
11:     Update the weight of each expert:

$$\theta_e^{t+1} = \frac{\theta_e^t \exp\left\{-\gamma \hat{l}_t(\hat{p}_e^t; \hat{p}^t)\right\}}{\sum_{e=1}^E \theta_e^t \exp\left\{-\gamma \hat{l}_t(\hat{p}_e^t; \hat{p}^t)\right\}}, \quad e \in [E].$$

12: **end for**

---

exponentially-weighted-average strategy to aggregate $\{\hat{p}_e^{1:T}\}_{e=1}^E$ into a single output $\hat{p}^{1:T}$, which achieves performance close to the best expert.

Algorithm 4 details Adaptive-OSMD Sampler. Note that since we can compute $\hat{l}_t(\hat{p}_e^t; \hat{p}^t)$ and $\nabla \hat{l}_t(\hat{p}_e^t; \hat{p}^t)$ directly, there is no need to use a surrogate loss as in van Erven & Koolen (2016) and Zhang et al. (2018).

From the computational perspective, the major cost comes from solving step 9 of Algorithm 4, which needs to be run for a total number of $T|\mathcal{E}| = T\lfloor \log_2 T \rfloor$ times. Compared with Algorithm 1, the computational complexity only increases by a $\log(T)$ factor. We have the following result on Algorithm 4.

**Theorem 3.** *Let $\Phi = \Phi_e$ and we use $\|\cdot\| = \|\cdot\|_1$ to define total variation. Let $\alpha$ be small enough such that $p_\star^t \in \mathcal{A}$ for all $t \in [T]$. Let $\hat{p}^{1:T}$ be the output of Algorithm 4 with $\gamma = \frac{\alpha}{M}\sqrt{\frac{8K}{T\mathbb{E}[\bar{a}^1]}}$, $p^{\text{init}} = p^{\text{unif}}$ and $\mathcal{E}$ as in equation 13. Then*

$$D\text{-}Regret_T \leq \frac{3\sqrt{2}M^3\sqrt{\mathbb{E}[(\bar{a}^1)^2]}}{K^2\alpha^3}\sqrt{T\left[\log M + 2\log(M/\alpha)\,\mathbb{E}\left[TV(p_\star^{1:T})\right]\right]}$$

$$+ \frac{M}{\alpha}\sqrt{\frac{T\mathbb{E}[\bar{a}^1]}{8K}}(1 + 2\log E).$$

*Proof.* See Appendix D.4. □

Since the additional regret term is $\tilde{O}((M/\alpha)\sqrt{T/K})$, which is no larger than the first term asymptotically, the bound on the regret is of the same order as in equation 12. However, we do not need to know the total variation $\mathbb{E}[TV(p_\star^{1:T})]$ to set the learning rate.

Based on Theorem 3, the choice of $\alpha$ relies on prior knowledge about $\{p_\star^t\}_{t \geq 1}$. Specifically, we need $\alpha$ to be small enough so that $p_\star^t \in \mathcal{A}$ for all $t \in [T]$. While this prior knowledge is not generally available, in Section F.3 we experimentally show that the proposed algorithm is robust to the choice of $\alpha$. As long as the chosen $\alpha$ is not too small or too large, we obtain a reasonable solution. We always set $\alpha = 0.4$ in experiments.

## C.4 Adaptive-OSMD Sampler with Doubling Trick (Adaptive-Doubling-OSMD)

Algorithm 4 requires the total number of iterations $T$ as input, which is not always available in practice. In those cases, we use doubling trick (Cesa-Bianchi & Lugosi, 2006, Section 2.3) to avoid this requirement. The basic idea is to restart Adaptive-OSMD Sampler at exponentially increasing time points $T_b = 2^{b-1}$, $b \geq 1$. The learning rates of experts in Algorithm 4 are reset at the beginning of each time interval, and the meta-algorithm learning rate $\gamma$ is chosen optimally for the interval length. We set $\bar{a}^{T_b}$ at the beginning of each time interval using the maximum environment feedback from the previous interval.

More specifically, at the time point $T_b$, we let

$$\mathcal{E}_b := \left\{ 2^{e-\frac{b}{2}-1} \cdot \frac{K^2 \alpha^3 \sqrt{\log M}}{M^3 \hat{a}^b} \,\middle|\, e = 1, 2, \ldots, E_b \right\}, \tag{15}$$

where

$$E_b = \lfloor \frac{1}{2} \log_2 \left( 1 + \frac{4 \log(M/\alpha)}{\log M}(2^{b-1} - 1) \right) \rfloor + 1, \tag{16}$$

and $\gamma_b = \frac{\alpha}{M}\sqrt{\frac{8K}{2^{b-1}\hat{a}^b}}$, $b \geq 1$. We set $\hat{a}^{T_b} = \max_{m \in S^{T_b-1}} a_m^{T_b-1}$. In a practical implementation, at the time point $t = T_b$, instead of initializing all expert algorithms using uniform distribution, we can initialize them with the output of the meta-algorithm for $t = T_b - 1$. To get $a^0$, the server uses a pre-training phase where the initial model parameter $w^0$ is broadcast to all devices before the start of the training. Subsequently, the server collects the returned $\|g_m^0\|^2$ from all responsive devices, which are denoted as $S^0$. Then $\hat{a}^1 := \max_{m \in S^0} a_m^0$ where $a_m^0 = \lambda_m \|g_m^0\|^2$. Adaptive-Doubling-OSMD Sampler is detailed in Algorithm 5.

From the computational perspective, by the proof of Theorem 4, Algorithm 5 needs to run Step 9 of Algorithm 4 for a total number of $O(T|\mathcal{E}|^2) = O(T\lfloor \log_2 T \rfloor)$ times. Therefore, the computational complexity of Adaptive-Doubling-OSMD Sampler is asymptotically the same as that of Adaptive-OSMD Sampler, while it increases by only a $\log(T)$ factor compared to OSMD Sampler. The following theorem provides a bound on the dynamic regret for Adaptive-Doubling-OSMD Sampler.

**Theorem 4.** *Let $\Phi = \Phi_e$ and we use $\|\cdot\| = \|\cdot\|_1$ to define the total variation. Let $\alpha$ be small enough constant so that $p_\star^{1:T} \in \mathcal{A}^T$ and the training is stopped after $T$ iterations. Suppose that there exists a constant $C > 1$ such that $\hat{a}^{T_b} \leq C\bar{a}^{T_b}$ for all $b = 1, 2, \ldots, B$, where $B = \lfloor \log_2(T+1) \rfloor$. Let $\hat{p}^{1:T}$ be the output of Algorithm 5, where $p^{\text{unif}}$ is used in Step 8. Then*

$$\text{D-Regret}_T$$
$$\leq \frac{\sqrt{2(T+1)}}{\sqrt{2}-1} \left\{ \frac{(2C+1)\sqrt{2}M^3\sqrt{\mathbb{E}\left[(\bar{a}^1)^2\right]}}{K^2\alpha^3} \sqrt{\log M + 2\log(M/\alpha)\,\mathbb{E}\left[TV(p_\star^{1:T})\right]} + \right.$$
$$\left. \frac{M}{\alpha}\sqrt{\frac{\mathbb{E}\left[\bar{a}^1\right]}{8K}}(C + 2\log E) \right\}.$$

*Proof.* See Appendix D.5. □

From Theorem 4 we observe that the asymptotic regret bound has the same order as that of OSMD Sampler and Adaptive-OSMD Sampler. However, Adaptive-Doubling-OSMD Sampler does not need to know $\mathbb{E}\left[TV(p_\star^{1:T})\right]$ or $T$ in advance.

## C.5 Adaptive Sampling Without Replacement

In the discussion so far, we have assumed that the set $S^t$ is obtained by sampling with replacement from $p^t$. However, when $K$ is relatively large compared to $M$ and $p^t$ is far from uniform distribution, sampling without replacement can be more efficient than sampling with replacement. However, when sampling without replacement using $p^t$, the variance reduction loss does not have a clean form as in equation 4. As a result, an online design of the sampling distribution is more challenging. In this section, we discuss how to use the sampling distribution obtained by Adaptive-OSMD Sampler to sample clients without replacement, following the approach taken in Hanchi & Stephens (2020).

---

**Algorithm 5** Adaptive-OSMD Sampler with Doubling Trick (Adaptive-Doubling-OSMD)

---

1: **Input:** Paramter $\alpha$.
2: **Output:** $\hat{p}^t$ for $t = 1, \ldots, T$.
3: Use $w^0$ to get $\{a_m^0\}_{m \in S^0}$, where $S^0$ is the set of responsive clients in the pre-training phase.
4: Initialize $\hat{\tilde{a}}^1 = \max_{m \in S^0} a_m^0$ and set $b = 1$.
5: **while** True **do**
6:     Set $\mathcal{E}_b$ as in equation 15.
7:     Let $\gamma_b = \frac{\alpha}{M} \sqrt{\frac{8K}{2^{b-1}\hat{\tilde{a}}^b}}$.
8:     Obtain $\{\hat{p}^t\}_{t=2^{b-1}}^{2^b-1}$ from Algorithm 4 with parameters: $\gamma_b$, $\mathcal{E}_b$, $\alpha$, the number of iterations $2^{b-1}$, and the initial distribution $p^{\text{unif}}$ or $\hat{p}^{2^{b-1}-1}$ (when $b > 1$).
9:     **if** Training Process is Converged **then**
10:         Break.
11:     **end if**
12:     Let $b \leftarrow b + 1$.
13:     Let $\hat{\tilde{a}}^b = \max_{m \in [M]} a_m^{T_b}$.
14: **end while**

---

**Algorithm 6** Adaptive sampling without replacement

---

1: **Input:** $w^1$ and $\hat{p}^1$.
2: **for** $t = 1, 2, \ldots, T - 1$ **do**
3:     Let $\hat{p}_{(1)}^t = \hat{p}^t$ and sample $m_1^t$ from $[M]$ by $\hat{p}_{(1)}^t$.
4:     **for** $k = 2, \cdots, K$ **do**
5:         /* Design the sampling distribution for sampling the $k$-th client in the $t$-th round */
6:         Construct $\hat{p}_{(k)}^t$ by letting

$$\hat{p}_{(k),m}^t = \begin{cases} \left(1 - \sum_{l=1}^{k-1} \hat{p}_{m_l^t}^t\right)^{-1} \hat{p}_m^t & \text{if } m \in [M] \backslash \{m_1^t, \ldots, m_{k-1}^t\} \\ 0 & \text{otherwise.} \end{cases}$$

7:         /* Sample the $k$-th client */
8:         Sample $m_k^t$ from $[M] \backslash \{m_1^t, \ldots, m_{k-1}^t\}$ by $\hat{p}_{(k)}^t$.
9:     **end for**
10:     Let $S^t = \{m_1^t, \cdots, m_K^t\}$.
11:     The server broadcasts the model parameter $w^t$ to clients in $S^t$.
12:     The clients in $S^t$ compute and upload the set of local gradients $\left\{g_{m_1^t}^t, \cdots, g_{m_K^t}^t\right\}$.
13:     /* Construct global gradient estimate */
14:     Let $g_{(1)}^t = \lambda_{m_1^t}^t g_{m_1^t}^t / \hat{p}_{(1),m_1^t}^t$.
15:     **for** $k = 2, \cdots, K$ **do**
16:         Let $g_{(k)}^t = \lambda_{m_k^t}^t g_{m_k^t}^t / \hat{p}_{(k),m_k^t}^t + \sum_{l=1}^{k-1} \lambda_{m_l^t}^t g_{m_l^t}^t$.
17:     **end for**
18:     Let $\tilde{g}^t = K^{-1} \sum_{k=1}^K g_{(k)}^t$.
19:     /* Update the model weight based on the global gradient estimate */
20:     Obtain the updated model parameter $w^{t+1}$ using $w^t$ and $\tilde{g}^t$.
21:     /* Update sampling distribution */
22:     Let $a_m^t = \lambda_m^2 \|g_m^t\|^2$ for $m \in S^t$.
23:     Input $\{a_m^t\}_{m \in S^t}$ into Adaptive-OSMD Sampler to get $\hat{p}^{t+1}$.
24: **end for**

---

The detailed sampling procedure is described in Algorithm 6. We still use Adaptive-OSMD Sampler to update the sampling distribution. However, we use the designed sampling distribution in a way that no client is chosen twice. Furthermore, Step 18 of Algorithm 6 constructs the gradient estimate with the following properties.

**Proposition 2** (Proposition 3 of Hanchi & Stephens (2020))**.** *Let $\hat{p}^t = p$ and let $\tilde{g}^t$ be as in Step 18 of Algorithm 6. Note that $\tilde{g}^t = \tilde{g}^t(p)$ depends on $p$. Recall that $J^t = \sum_{m=1}^{M} \lambda_m g_m^t$. We have*

$$\mathbb{E}_{S^t}\left[\tilde{g}^t\right] = J^t \qquad and \qquad \arg\min_{p \in \mathcal{P}_{M-1}} \mathbb{E}_{S^t}\left[\|\tilde{g}^t - J^t\|_2^2\right] = \arg\min_{p \in \mathcal{P}_{M-1}} l_t(p),$$

*where $l_t(\cdot)$ is defined in equation 4 and the expectation is taken over $S^t$.*

From Proposition 2, we see that $\tilde{g}^t$ is an unbiased stochastic gradient. Furthermore, the variance of $\tilde{g}^t$ is minimized by the same sampling distribution that minimizes the variance reduction loss in equation 4. Therefore, it is reasonable to use the sampling distribution generated by Adaptive-OSMD Sampler to design $\tilde{g}^t$.

## D  TECHNICAL PROOFS

### D.1  PROOF OF PROPOSITION 1

First, we show that the solution $\hat{p}^{t+1}$ in Step 7 of Algorithm 1 can be found as

$$\tilde{p}^{t+1} = \arg\min_{p \in \mathcal{D}} \eta_t \langle p, \nabla \hat{l}_t(\hat{p}^t; \hat{p}^t) \rangle + D_{\Phi_e}\left(p \,\|\, \hat{p}^t\right),$$

$$\hat{p}^{t+1} = \arg\min_{p \in \mathcal{A}} D_{\Phi_e}\left(p \,\|\, \tilde{p}^{t+1}\right).$$

The optimality condition for $\tilde{p}^{t+1}$ implies that

$$\eta_t \nabla \hat{l}_t(\hat{p}^t; \hat{p}^t) + \nabla \Phi_e(\tilde{p}^{t+1}) - \nabla \Phi_e(\hat{p}^t) = 0. \tag{17}$$

By Lemma 1, the optimality condition for $\hat{p}^{t+1}$ implies that

$$\langle p - \hat{p}^{t+1}, \nabla \Phi_e(\hat{p}^{t+1}) - \nabla \Phi_e(\tilde{p}^{t+1}) \rangle \geq 0, \quad \text{for all } p \in \mathcal{A}.$$

Combining the last two displays, we have

$$\langle p - \hat{p}^{t+1}, \eta_t \nabla \hat{l}_t(\hat{p}^t; \hat{p}^t) + \nabla \Phi_e(\hat{p}^{t+1}) - \nabla \Phi_e(\hat{p}^t) \rangle \geq 0, \quad \text{for all } p \in \mathcal{A}.$$

By Lemma 1, this is the optimality condition for $\hat{p}^{t+1}$ to be the solution in Step 7 of Algorithm 1.

Note that equation 17 implies that

$$-\frac{\eta_t}{K^2} \cdot \frac{a_m^t}{(p_m^t)^3} \mathcal{N}\left\{m \in S^t\right\} + \log(\tilde{p}_m^{t+1}) - \log(\hat{p}_m^t) = 0, \qquad m \in [M].$$

Therefore,

$$\tilde{p}_m^{t+1} = \hat{p}_m^t \exp\left(\frac{\eta_t a_m^t}{K^2 \left(\hat{p}_m^t\right)^3} \mathcal{N}\left\{m \in S^t\right\}\right), \qquad m \in [M],$$

and the final result follows from Lemma 4.

### D.2  PROOF OF THEOREM 1

We first state a proposition that will be used to prove Theorem 1. The key difference between Theorem 1 and Proposition 3 is that in Proposition 3 the comparator sequence lies in $\mathcal{A}$, and, as a result, there is no projection error.

**Proposition 3.** *Suppose the conditions of Theorem 1 hold. For any comparator sequence $q^{1:T}$ with $q^t \in \mathcal{A}$, $t \in [T]$, we have*

$$D\text{-}Regret_T(q^{1:T}) \leq \frac{D_{\max}}{\eta_1} + \frac{2H}{\eta_T}\mathbb{E}\left[TV\left(q^{1:T}\right)\right] + \frac{2}{\rho}\sum_{t=1}^{T} \eta_t \mathbb{E}\left[Q_t^2\right].$$

*Proof.* By Lemma 1 and the definition of $\hat{p}^{t+1}$ in Step 7 of Algorithm 1, we have

$$\langle \hat{p}^{t+1} - q^t, \nabla \hat{l}_t(\hat{p}^t; \hat{p}^t) \rangle \leq \frac{1}{\eta_t} \langle \nabla \Phi(\hat{p}^t) - \nabla \Phi(\hat{p}^{t+1}), \hat{p}^{t+1} - q^t \rangle. \tag{18}$$

By the convexity of $\hat{l}_t(\cdot; \hat{p}^t)$, we have

$$\hat{l}_t(\hat{p}^t; \hat{p}^t) - \hat{l}_t(q^t; \hat{p}^t) \leq \langle \nabla \hat{l}_t(\hat{p}^t; \hat{p}^t), \hat{p}^t - q^t \rangle = \langle \nabla \hat{l}_t(\hat{p}^t; \hat{p}^t), \hat{p}^{t+1} - q^t \rangle + \langle \nabla \hat{l}_t(\hat{p}^t; \hat{p}^t), \hat{p}^t - \hat{p}^{t+1} \rangle.$$

Then, by equation 18, we further have

$$\hat{l}_t(\hat{p}^t; \hat{p}^t) - \hat{l}_t(q^t; \hat{p}^t) \leq \frac{1}{\eta_t} \langle \nabla \Phi(\hat{p}^t) - \nabla \Phi(\hat{p}^{t+1}), \hat{p}^{t+1} - q^t \rangle + \langle \nabla \hat{l}_t(\hat{p}^t; \hat{p}^t), \hat{p}^t - \hat{p}^{t+1} \rangle.$$

From the definition of $\mathcal{D}$, we have

$$D_\Phi(x_1 \| x_2) = D_\Phi(x_3 \| x_2) + D_\Phi(x_1 \| x_3) + \langle \nabla \Phi(x_2) - \nabla \Phi(x_3), x_3 - x_1 \rangle, \qquad x_1, x_2, x_3 \in \mathcal{D}.$$

Then

$$\begin{aligned}
\hat{l}_t(\hat{p}^t; \hat{p}^t) &- \hat{l}_t(q^t; \hat{p}^t) \\
&\leq \frac{1}{\eta_t} \left[ D_\Phi(q^t \| \hat{p}^t) - D_\Phi(q^t \| \hat{p}^{t+1}) - D_\Phi(\hat{p}^{t+1} \| \hat{p}^t) \right] + \langle \nabla \hat{l}_t(\hat{p}^t; \hat{p}^t), \hat{p}^t - \hat{p}^{t+1} \rangle \\
&= \frac{1}{\eta_t} \left[ D_\Phi(q^t \| \hat{p}^t) - D_\Phi(q^{t+1} \| \hat{p}^{t+1}) \right] + \frac{1}{\eta_t} \left[ D_\Phi(q^{t+1} \| \hat{p}^{t+1}) - D_\Phi(q^t \| \hat{p}^{t+1}) \right] \\
&\quad - \frac{1}{\eta_t} D_\Phi(\hat{p}^{t+1} \| \hat{p}^t) + \langle \nabla \hat{l}_t(\hat{p}^t; \hat{p}^t), \hat{p}^t - \hat{p}^{t+1} \rangle.
\end{aligned} \tag{19}$$

We bound the second term in equation 19 as

$$\begin{aligned}
D_\Phi(q^{t+1} \| \hat{p}^{t+1}) - D_\Phi(q^t \| \hat{p}^{t+1}) &= \Phi(q^{t+1}) - \Phi(q^t) - \langle \nabla \Phi(\hat{p}^{t+1}), q^{t+1} - q^t \rangle \\
&\overset{(a)}{\leq} \langle \nabla \Phi(q^{t+1}) - \nabla \Phi(\hat{p}^{t+1}), q^{t+1} - q^t \rangle \\
&\overset{(b)}{\leq} \| \nabla \Phi(q^{t+1}) - \nabla \Phi(\hat{p}^{t+1}) \|_\star \| q^{t+1} - q^t \| \\
&\overset{(c)}{\leq} 2H \| q^{t+1} - q^t \|,
\end{aligned} \tag{20}$$

where (a) follows from the convexity of $\Phi(\cdot)$, (b) follows from the definition of the dual norm, and (c) follows from the definition of $H$. Since $\Phi(\cdot)$ is $\rho$-strongly convex, we can bound the third and fourth term in equation 19 as

$$-\frac{1}{\eta_t} D_\Phi(\hat{p}^{t+1} \| \hat{p}^t) + \langle \nabla \hat{l}_t(\hat{p}^t; \hat{p}^t), \hat{p}^t - \hat{p}^{t+1} \rangle \leq -\frac{\rho}{2\eta_t} \| \hat{p}^{t+1} - \hat{p}^t \|^2 + \| \nabla \hat{l}_t(\hat{p}^t; \hat{p}^t) \|_\star \| \hat{p}^t - \hat{p}^{t+1} \|.$$

Since $ab \leq a^2/(2\epsilon) + b^2 \epsilon/2$, $a, b, \epsilon > 0$, we further have

$$-\frac{1}{\eta_t} D_\Phi(\hat{p}^{t+1} \| \hat{p}^t) + \langle \nabla \hat{l}_t(\hat{p}^t; \hat{p}^t), \hat{p}^t - \hat{p}^{t+1} \rangle \leq \frac{2\eta_t}{\rho} \| \nabla \hat{l}_t(\hat{p}^t; \hat{p}^t) \|_\star^2 \leq \frac{2\eta_t}{\rho} Q_t^2. \tag{21}$$

Combining equation 19-equation 21, we have

$$\hat{l}_t(\hat{p}^t; \hat{p}^t) - \hat{l}_t(q^t; \hat{p}^t) \leq \frac{D_\Phi(q^t \| \hat{p}^t)}{\eta_t} - \frac{D_\Phi(q^{t+1} \| \hat{p}^{t+1})}{\eta_t} + 2H \frac{\| q^{t+1} - q^t \|}{\eta_t} + \frac{2}{\rho} \eta_t Q_t^2.$$

This implies that

$$\begin{aligned}
\sum_{t=1}^T \hat{l}_t(\hat{p}^t; \hat{p}^t) &- \sum_{t=1}^T \hat{l}_t(q^t; \hat{p}^t) \\
&\leq \frac{D_\Phi(q^1 \| \hat{p}^1)}{\eta_1} - \frac{D_\Phi(q^{T+1} \| \hat{p}^{T+1})}{\eta_{T+1}} + 2H \sum_{t=1}^T \frac{\| q^{t+1} - q^t \|}{\eta_t} + \frac{2}{\rho} \sum_{t=1}^T \eta_t Q_t^2 \\
&\leq \frac{D_\Phi(q^1 \| \hat{p}^1)}{\eta_1} + \frac{2H}{\eta_T} \sum_{t=1}^T \| q^{t+1} - q^t \| + \frac{2}{\rho} \sum_{t=1}^T \eta_t Q_t^2 \\
&\leq \frac{D_{\max}}{\eta_1} + \frac{2H}{\eta_T} \text{TV}(q^{1:T}) + \frac{2}{\rho} \sum_{t=1}^T \eta_t Q_t^2,
\end{aligned} \tag{22}$$

since $\hat{p}_1$ is the uniform distribution. Finally, note that

$$
\mathbb{E}\left[\sum_{t=1}^{T} \hat{l}_t(\hat{p}^t; \hat{p}^t) - \sum_{t=1}^{T} \hat{l}_t(q^t; \hat{p}^t)\right] = \sum_{t=1}^{T} \mathbb{E}\left[\mathbb{E}_{S^t}\left[\hat{l}_t(\hat{p}^t; \hat{p}^t)\right] - \mathbb{E}_{S^t}\left[\hat{l}_t(q^t; \hat{p}^t)\right]\right]
$$

$$
= \sum_{t=1}^{T} \mathbb{E}\left[l_t(\hat{p}^t) - l_t(q^t)\right]
$$

$$
= \text{D-Regret}_T(q^{1:T}).
$$

The conclusion follows by taking expectation on both sides of equation 22. □

We are now ready to prove Theorem 1.

*Proof of Theorem 1.* For any comparator sequence $q^{1:T}$ with $q^t \in \mathcal{P}_{M-1}$, $t \in [T]$, we have

$$
\text{D-Regret}_T(q^{1:T}) = \mathbb{E}\left[\sum_{t=1}^{T} l_t(\hat{p}^t) - \sum_{t=1}^{T} l_t(\tilde{q}^t) + \sum_{t=1}^{T} l_t(\tilde{q}^t) - \sum_{t=1}^{T} l_t(q^t)\right]. \tag{23}
$$

By Proposition 3, we further have that

$$
\mathbb{E}\left[\sum_{t=1}^{T} l_t(\hat{p}^t) - \sum_{t=1}^{T} l_t(\tilde{q}^t)\right]
$$

$$
\leq \frac{D_{\max}}{\eta_1} + \frac{2H}{\eta_T}\mathbb{E}\left[\text{TV}\left(q^{1:T}\right)\right] + \frac{2}{\rho}\sum_{t=1}^{T} \eta_t \mathbb{E}\left[Q_t^2\right] + \frac{2H}{\eta_T}\mathbb{E}\left[\text{TV}\left(\tilde{q}^{1:T}\right) - \text{TV}\left(q^{1:T}\right)\right]. \tag{24}
$$

Therefore, to prove Theorem 1, we design a suitable sequence $\tilde{q}^{1:T}$, where $\tilde{q}^t \in \mathcal{A}$, and bound the terms $\sum_{t=1}^{T} l_t(\tilde{q}^t) - \sum_{t=1}^{T} l_t(q^t)$ and $\text{TV}\left(\tilde{q}^{1:T}\right) - \text{TV}\left(q^{1:T}\right)$.

We define $\tilde{q}^t$ as

$$
\tilde{q}_m^t = \begin{cases} \alpha/M & \text{if } q_m^t < \alpha/M, \\ q_m^t - \omega(q^t, \alpha)\left(q_m^t - \frac{\alpha}{M}\right) & \text{if } q_m^t \geq \alpha/M, \end{cases} \tag{25}
$$

where $\omega(q^t, \alpha)$ is defined in equation 9. We now show that $\tilde{q}^t \in \mathcal{A}$, $t \in [T]$, by showing that $\tilde{q}_m^t \geq \alpha/M$, $m \in [M]$, and $\sum_{m \in [M]} \tilde{q}_m^t = 1$. For $m \in [M]$ such that $q_m^t < \alpha/M$, we have from equation 25 that $\tilde{q}_m^t = \alpha/M$. For $m \in [M]$ such that $q_m^t \geq \alpha/M$, by equation 25, we have $\tilde{q}_m^t - \alpha/M = (1 - \omega(q^t, \alpha))\left(q_m^t - \alpha/M\right)$. Thus, we proceed to show that $\omega(q^t, \alpha) \leq 1$. Since

$$
1 = \sum_{m=1}^{M} q_m^t \mathbb{1}\left\{q_m^t < \frac{\alpha}{M}\right\} + \sum_{m=1}^{M} q_m^t \mathbb{1}\left\{q_m^t \geq \frac{\alpha}{M}\right\}
$$

$$
\geq \sum_{m=1}^{M} \frac{\alpha}{M}\mathbb{1}\left\{q_m^t < \frac{\alpha}{M}\right\} + \sum_{m=1}^{M} \frac{\alpha}{M}\mathbb{1}\left\{q_m^t \geq \frac{\alpha}{M}\right\} = \alpha,
$$

we have

$$
\sum_{m=1}^{M}\left(q_m^t - \frac{\alpha}{M}\right)\mathbb{1}\left\{q_m^t \geq \frac{\alpha}{M}\right\} \geq \sum_{m=1}^{M}\left(\frac{\alpha}{M} - q_m^t\right)\mathbb{1}\left\{q_m^t < \frac{\alpha}{M}\right\}.
$$

Therefore, $0 \le \omega(q^t, \alpha) \le 1$. Furthermore, $\omega(q^t, 0) = 1$ and $\omega(q^t, 1) = 1$. Finally, we show that $\sum_{m=1}^{M} \tilde{q}_m^t = 1$. By equation 25 and the definition of $\omega(q^t, \alpha)$ in equation 9, we have

$$
\begin{aligned}
\sum_{m=1}^{M} \tilde{q}_m^t &= \sum_{m=1}^{M} \frac{\alpha}{M} \mathbb{1}\left\{q_m^t < \frac{\alpha}{M}\right\} + \sum_{m=1}^{M} q_m^t \mathbb{1}\left\{q_m^t \ge \frac{\alpha}{M}\right\} \\
&\quad - \omega(q^t, \alpha) \sum_{m=1}^{M} \left(q_m^t - \frac{\alpha}{M}\right) \mathbb{1}\left\{q_m^t \ge \frac{\alpha}{M}\right\} \\
&= \sum_{m=1}^{M} \frac{\alpha}{M} \mathbb{1}\left\{q_m^t < \frac{\alpha}{M}\right\} + \sum_{m=1}^{M} q_m^t \mathbb{1}\left\{q_m^t \ge \frac{\alpha}{M}\right\} - \sum_{m=1}^{M} \left(\frac{\alpha}{M} - q_m^t\right) \mathbb{1}\left\{q_m^t < \frac{\alpha}{M}\right\} \\
&= \sum_{m=1}^{M} q_m^t \mathbb{1}\left\{q_m^t \ge \frac{\alpha}{M}\right\} + \sum_{m=1}^{M} q_m^t \mathbb{1}\left\{q_m^t < \frac{\alpha}{M}\right\} \\
&= 1.
\end{aligned}
$$

Therefore, $\tilde{q}^t \in \mathcal{A}$ for any $t \in [T]$.

We now bound $\sum_{t=1}^{T} l_t(\tilde{q}^t) - \sum_{t=1}^{T} l_t(q^t)$. When $q_m^t < \alpha/M$, then $1/\tilde{q}_m^t - 1/q_m^t < 0$; and when $q_m^t \ge \alpha/M$, then

$$
\frac{1}{\tilde{q}_m^t} - \frac{1}{q_m^t} = \frac{1}{q_m^t} \cdot \left[\frac{1}{1 - \omega(q^t, \alpha)\left(1 - \frac{\alpha}{Mq_m^t}\right)} - 1\right] = \frac{1}{q_m^t} \cdot \frac{\omega(q^t, \alpha)\left(1 - \frac{\alpha}{Mq_m^t}\right)}{1 - \omega(q^t, \alpha)\left(1 - \frac{\alpha}{Mq_m^t}\right)}.
$$

Since

$$
\omega(q^t, \alpha)\left(1 - \frac{\alpha}{Mq_m^t}\right) \le \omega(q^t, \alpha) \quad \text{and} \quad 1 - \omega(q^t, \alpha)\left(1 - \frac{\alpha}{Mq_m^t}\right) \ge 1 - \omega(q^t, \alpha) + \frac{\omega(q^t, \alpha)\alpha}{M}
$$

as $q_m^t \le 1$, we have

$$
\frac{1}{\tilde{q}_m^t} - \frac{1}{q_m^t} \le \frac{1}{q_m^t} \cdot \frac{\omega(q^t, \alpha)}{1 - \omega(q^t, \alpha)\left(1 - \frac{\alpha}{M}\right)} = \frac{\phi(q^t, \alpha)}{q_m^t}.
$$

Thus,

$$
\begin{aligned}
\sum_{t=1}^{T} l_t(\tilde{q}^t) - \sum_{t=1}^{T} l_t(q^t) &= \sum_{t=1}^{T} \sum_{m=1}^{M} a_m^t \left(\frac{1}{\tilde{q}_m^t} - \frac{1}{q_m^t}\right) \\
&\le \sum_{t=1}^{T} \sum_{m=1}^{M} a_m^t \left(\frac{1}{\tilde{q}_m^t} - \frac{1}{q_m^t}\right) \mathbb{1}\left\{q_m^t \ge \frac{\alpha}{M}\right\} \\
&\le \sum_{t=1}^{T} \phi(q^t, \alpha) \sum_{m=1}^{M} \frac{a_m^t}{q_m^t} \mathbb{1}\left\{q_m^t \ge \frac{\alpha}{M}\right\} \\
&\le \sum_{t=1}^{T} \phi(q^t, \alpha) l_t(q^t).
\end{aligned} \tag{26}
$$

Next, we bound $\text{TV}\left(\tilde{q}^{1:T}\right) - \text{TV}\left(q^{1:T}\right)$. Note that

$$
\begin{aligned}
\text{TV}\left(\tilde{q}^{1:T}\right) &= \sum_{t=2}^{T}\left\|\tilde{q}^{t} - \tilde{q}^{t-1}\right\|_{1} \\
&= \sum_{t=2}^{T}\left\|\tilde{q}^{t} - q^{t} + q^{t} - q^{t-1} + q^{t-1} - \tilde{q}^{t-1}\right\|_{1} \\
&\leq \sum_{t=2}^{T}\left\|\tilde{q}^{t} - q^{t}\right\|_{1} + \sum_{t=2}^{T}\left\|q^{t} - q^{t-1}\right\|_{1} + \sum_{t=2}^{T}\left\|q^{t-1} - \tilde{q}^{t-1}\right\|_{1} \\
&\leq \text{TV}\left(q^{1:T}\right) + 2\sum_{t=1}^{T}\left\|\tilde{q}^{t} - q^{t}\right\|_{1}.
\end{aligned}
$$

We now upper bound $\sum_{t=1}^{T}\left\|\tilde{q}^{t} - q^{t}\right\|_{1}$. If $q_{m}^{t} < \alpha/M$, then $|\tilde{q}_{m}^{t} - q_{m}^{t}| = \alpha/M - q_{m}^{t}$. If $q_{m}^{t} \geq \alpha/M$, by equation 25, we have $|\tilde{q}_{m}^{t} - q_{m}^{t}| = \omega(q^{t}, \alpha)\left(q_{m}^{t} - \alpha/M\right)$. Therefore, recalling the definition of $\psi(q^{t}, \alpha)$ in equation 9, we have

$$
\begin{aligned}
\left\|\tilde{q}^{t} - q^{t}\right\|_{1} &= \sum_{m=1}^{M}\left(\frac{\alpha}{M} - q_{m}^{t}\right)\mathbb{1}\left\{q_{m}^{t} < \frac{\alpha}{M}\right\} + \omega(q^{t}, \alpha)\sum_{m=1}^{M}\left(q_{m}^{t} - \frac{\alpha}{M}\right)\mathbb{1}\left\{q_{m}^{t} \geq \frac{\alpha}{M}\right\} \\
&= 2\sum_{m=1}^{M}\left(\frac{\alpha}{M} - q_{m}^{t}\right)\mathbb{1}\left\{q_{m}^{t} < \frac{\alpha}{M}\right\} \\
&= 2\psi(q^{t}, \alpha)
\end{aligned}
$$

and

$$
\text{TV}\left(\tilde{q}^{1:T}\right) - \text{TV}\left(q^{1:T}\right) \leq 4\sum_{t=1}^{T}\psi(q^{t}, \alpha). \tag{27}
$$

Combining equation 23, equation 24, equation 26, and equation 27, and taking expectation on both sides, we obtain the result. $\qquad\square$

## D.3 COROLLARY 3 AND ITS PROOF

**Corollary 3.** *Suppose conditions of Theorem 1 hold and $\Phi = \Phi_{e}$. Then for any comparator sequence $\{q^{1:T}\}$, we have*

$$
\begin{aligned}
\textit{D-Regret}_{T}(q^{1:T}) \leq \frac{\log M}{\eta_{1}} &+ \frac{2\log(M/\alpha)}{\eta_{T}}\textit{TV}\left(q^{1:T}\right) + \frac{2M^{6}}{K^{4}\alpha^{6}}\sum_{t=1}^{T}\eta_{t}\left(\bar{a}^{t}\right)^{2} \\
&+ \frac{8H}{\eta_{T}}\sum_{t=1}^{T}\psi(q^{t}, \alpha) + \sum_{t=1}^{T}\phi(q^{t}, \alpha)l_{t}(q^{t}),
\end{aligned}
$$

*where $\bar{a}^{t} \coloneqq \max_{1\leq m\leq M}a_{m}^{t} = \max_{1\leq m\leq M}\lambda_{m}^{2}\|g_{m}^{t}\|^{2}$.*

*Proof.* When $\Phi = \Phi_{e}$, we have $\rho = 1$ from Pinsker's inequality. Furthermore, $\|\cdot\|_{\star} = \|\cdot\|_{\infty}$, $\nabla\Phi_{e}(p) = (\log p_{1}, \ldots, \log p_{M})^{\top}$, and $\hat{p}^{t} \in \mathcal{A} = \mathcal{P}_{M-1} \cap [\alpha/M, \infty)^{M}$, $t \in [T]$. Then $Q_{t} = M^{3}\bar{a}^{t}/(K^{2}\alpha^{3})$ and $H = \log(M/\alpha)$ follows by checking the definition. Finally, to show that $D_{\Phi_{e}}(q \,\|\, p^{\text{unif}}) \leq \log M$ for all $q \in \mathcal{A}$, we note that $D_{\Phi_{e}}(q \,\|\, p^{\text{unif}}) = \log M + \sum_{m=1}^{M}q_{m}\log q_{m} \leq \log M$. $\qquad\square$

## D.4 PROOF OF THEOREM 3

The proof proceeds in two steps. First, we show that there exists an expert learning rate $\eta_{e} \in \mathcal{E}$ such that the regret bound for $\hat{p}_{e}^{1:T}$ is close to equation 12. That is, we show that there exists $\eta_{e} \in \mathcal{E}$ such

that

$$\mathbb{E}\left[\sum_{t=1}^{T}\hat{l}_t(\hat{p}_e^t;\hat{p}^t) - \sum_{t=1}^{T}l_t(p_\star^t)\right] \leq \frac{3\sqrt{2}M^3\sqrt{\mathbb{E}\left[(\bar{a}^1)^2\right]}}{K^2\alpha^3}\sqrt{T\left[\log M + 2\log\left(M/\alpha\right)\mathbb{E}\left[\mathrm{TV}\left(p_\star^{1:T}\right)\right]\right]}.$$
(28)

Note that $S^t \sim \hat{p}^t$. Second, we show that the output of meta-algorithm can track the best expert with small regret. That is, we show that

$$\mathbb{E}\left[\sum_{t=1}^{T}l_t(\hat{p}^t)\right] - \mathbb{E}\left[\sum_{t=1}^{T}\hat{l}_t(\hat{p}_e^t;\hat{p}^t)\right] \leq \frac{M}{\alpha}\sqrt{\frac{T\mathbb{E}\left[\bar{a}^1\right]}{8K}}(1 + 2\log E), \qquad e \in [E]. \qquad (29)$$

The theorem follows by combining equation 28 and equation 29.

We first prove equation 28. Since $0 \leq \mathbb{E}[\mathrm{TV}\left(p_\star^{1:T}\right)] \leq 2(T-1)$, we have

$$\min\mathcal{E} = \frac{K^2\alpha^3}{M^3\bar{a}^1}\sqrt{\frac{\log M}{2T}} \leq \eta^\star \leq \frac{K^2\alpha^3}{M^3\bar{a}^1}\sqrt{\frac{\log M + 4\log(M/\alpha)(T-1)}{2T}} \leq \max\mathcal{E},$$

where $\eta^\star$ is defined as in equation 11. Thus, there exists $\eta_e \in \mathcal{E}$, such that $\eta_e \leq \eta^\star \leq 2\eta_e$. Repeating the proof of equation 22 and proof of Corollary 3, we show that

$$\sum_{t=1}^{T}\hat{l}_t(\hat{p}_e^t;\hat{p}^t) - \sum_{t=1}^{T}\hat{l}_t(p_\star^t;\hat{p}^t) \leq \frac{\log M}{\eta_e} + \frac{2\log(M/\alpha)}{\eta_e}\mathrm{TV}\left(p_\star^{1:T}\right) + \frac{2\eta_e M^6}{K^4\alpha^6}\sum_{t=1}^{T}\left(\bar{a}^t\right)^2,$$

which then implies that

$$\mathbb{E}\left[\sum_{t=1}^{T}\hat{l}_t(\hat{p}_e^t;\hat{p}^t) - \sum_{t=1}^{T}\hat{l}_t(p_\star^t;\hat{p}^t)\right] \leq \frac{\log M}{\eta_e} + \frac{2\log(M/\alpha)}{\eta_e}\mathbb{E}\left[\mathrm{TV}\left(p_\star^{1:T}\right)\right]$$

$$+ \frac{2\eta_e M^6}{K^4\alpha^6}\sum_{t=1}^{T}\mathbb{E}\left[\left(\bar{a}^t\right)^2\right],$$

Since $\eta^\star/2 \leq \eta_e \leq \eta^\star$, we further have

$$\mathbb{E}\left[\sum_{t=1}^{T}\hat{l}_t(\hat{p}_e^t;\hat{p}^t) - \sum_{t=1}^{T}\hat{l}_t(p_\star^t;\hat{p}^t)\right]$$

$$\leq \frac{2\log M}{\eta^\star} + \frac{4\log(M/\alpha)}{\eta^\star}\mathbb{E}\left[\mathrm{TV}\left(p_\star^{1:T}\right)\right] + \frac{2\eta^\star M^6}{K^4\alpha^6}\sum_{t=1}^{T}\mathbb{E}\left[\left(\bar{a}^t\right)^2\right]$$

$$\leq \frac{2\log M}{\eta^\star} + \frac{4\log(M/\alpha)}{\eta^\star}\mathbb{E}\left[\mathrm{TV}\left(p_\star^{1:T}\right)\right] + \frac{2\eta^\star M^6 T}{K^4\alpha^6}\mathbb{E}\left[\left(\bar{a}^1\right)^2\right]$$

$$= \frac{3\sqrt{2}M^3\sqrt{\mathbb{E}\left[(\bar{a}^1)^2\right]}}{K^2\alpha^3}\sqrt{T\left[\log M + 2\log\left(M/\alpha\right)\mathbb{E}\left[\mathrm{TV}\left(p_\star^{1:T}\right)\right]\right]}.$$

Now, equation 28 follows, since

$$\mathbb{E}\left[\sum_{t=1}^{T}\hat{l}_t(p_\star^t;\hat{p}^t)\right] = \sum_{t=1}^{T}\mathbb{E}\left[\mathbb{E}_{S^t}\left[\hat{l}_t(p_\star^t;\hat{p}^t)\right]\right] = \sum_{t=1}^{T}\mathbb{E}\left[l_t(p_\star^t)\right].$$

We prove equation 29 next. Let

$$\hat{L}_t^e = \sum_{s=1}^{t}\hat{l}_s(\hat{p}_e^s;\hat{p}^s) \quad e \in [E], t \in [T].$$

Recall the update for $\theta_e^t$ in Step 11 of Alg 4. We have

$$\theta_e^t = \frac{\theta_e^1\exp\left(-\gamma\hat{L}_{t-1}^e\right)}{\sum_{b=1}^{E}\theta_b^1\exp\left(-\gamma\hat{L}_{t-1}^b\right)}, \qquad t = 2,\ldots T.$$

Let $\Theta_t = \sum_{b=1}^{E} \theta_b^1 \exp\left\{-\gamma \hat{L}_t^b\right\}$. Then

$$\log \Theta_1 = \log \left(\sum_{b=1}^{E} \theta_b^1 \exp\left\{-\gamma \hat{L}_1^b\right\}\right)$$

and, for $t \geq 2$,

$$\log\left(\frac{\Theta_t}{\Theta_{t-1}}\right) = \log\left(\frac{\sum_{b=1}^{E} \theta_b^1 \exp\left\{-\gamma \hat{L}_{t-1}^b\right\} \exp\left\{-\gamma \hat{l}_t(\hat{p}_b^t; \hat{p}^t)\right\}}{\sum_{b=1}^{E} \theta_b^1 \exp\left\{-\gamma \hat{L}_{t-1}^b\right\}}\right)$$

$$= \log\left(\sum_{b=1}^{E} \theta_b^t \exp\left\{-\gamma \hat{l}_t(\hat{p}_b^t; \hat{p}^t)\right\}\right).$$

We have

$$\log \Theta_T = \log \Theta_1 + \sum_{t=1}^{T} \log\left(\frac{\Theta_t}{\Theta_{t-1}}\right)$$

$$= \sum_{t=1}^{T} \log\left(\sum_{b=1}^{E} \theta_b^t \exp\left\{-\gamma \hat{l}_t(\hat{p}_b^t; \hat{p}^t)\right\}\right)$$

$$\leq \sum_{t=1}^{T}\left(-\gamma \sum_{b=1}^{E} \theta_b^t \hat{l}_t(\hat{p}_b^t; \hat{p}^t) + \frac{\gamma^2 M^2 \bar{a}^t}{8K\alpha^2}\right) \qquad \text{(Lemma 3)}$$

$$\leq -\gamma \sum_{t=1}^{T} \hat{l}_t(\hat{p}^t; \hat{p}^t) + \frac{\gamma^2 M^2 \left(\sum_{t=1}^{T} \bar{a}^t\right)}{8K\alpha^2} \qquad \text{(Jensen's inequality)}$$

and

$$\log(\Theta_T) = \log\left(\sum_{b=1}^{E} \theta_b^1 \exp\left\{-\gamma \hat{L}_T^b\right\}\right)$$

$$\geq \log\left(\max_{1 \leq b \leq E} \theta_b^1 \exp\left\{-\gamma \hat{L}_T^b\right\}\right) = -\gamma \min_{1 \leq b \leq E}\left\{\hat{L}_T^b + \frac{1}{\gamma}\log\frac{1}{\theta_b^1}\right\}.$$

Combining the last two displays, we have

$$-\gamma \min_{1 \leq b \leq E}\left\{\hat{L}_T^b + \frac{1}{\gamma}\log\frac{1}{\theta_b^1}\right\} \leq -\gamma \sum_{t=1}^{T} \hat{l}_t(\hat{p}^t; \hat{p}^t) + \frac{\gamma^2 M^2 \left(\sum_{t=1}^{T} \bar{a}^t\right)}{8K\alpha^2},$$

which implies that

$$\sum_{t=1}^{T} \hat{l}_t(\hat{p}^t; \hat{p}^t) - \hat{L}_T^e \leq \frac{\gamma M^2 \left(\sum_{t=1}^{T} \bar{a}^t\right)}{8K\alpha^2} + \frac{1}{\gamma}\log\frac{1}{\theta_e^1} \leq \frac{\gamma M^2 T \bar{a}^1}{8K\alpha^2} + \frac{1}{\gamma}\log\frac{1}{\theta_e^1}, \quad e \in [E].$$

Taking expectation on both sides, we then have

$$\mathbb{E}\left[\sum_{t=1}^{T} \hat{l}_t(\hat{p}^t; \hat{p}^t) - \hat{L}_T^e\right] \leq \frac{\gamma M^2 T \mathbb{E}\left[\bar{a}^1\right]}{8K\alpha^2} + \frac{1}{\gamma}\log\frac{1}{\theta_e^1}$$

Since $\theta_e^1 \geq \frac{1}{E^2}$, $\log 1/\theta_e^1 \leq 2\log E$. Let $\gamma = \sqrt{8K\alpha^2/(TM^2\mathbb{E}[\bar{a}^1])}$ to minimize the right hand side of the above inequality with $\log 1/\theta_e^1$ substituted by 1. Then

$$\mathbb{E}\left[\sum_{t=1}^{T} \hat{l}_t(\hat{p}^t; \hat{p}^t) - \hat{L}_T^e\right] = \mathbb{E}\left[\sum_{t=1}^{T} \hat{l}_t(\hat{p}^t; \hat{p}^t) - \sum_{t=1}^{T} \hat{l}_t(\hat{p}_e^t; \hat{p}^t)\right]$$

$$\leq \frac{M}{\alpha}\sqrt{\frac{T\mathbb{E}\left[\bar{a}^1\right]}{8K}}\left(1 + 2\log E\right), \quad e \in [E].$$

### D.5 PROOF OF THEOREM 4

Recall that $T_b = 2^{b-1}$ and $\hat{p}^{T_b}$ is reinitialized as the uniform distribution. Let

$$\text{D-Regret}_b = \mathbb{E}\left[\sum_{t=T_b}^{T_{b+1}-1} l_t(\hat{p}^t) - \sum_{t=T_b}^{T_{b+1}-1} l_t(p_\star^t)\right].$$

Similar to the proof of equation 28 and equation 29, we have

$\text{D-Regret}_b$

$$\leq \frac{(2C+1)\sqrt{2}M^3\sqrt{\mathbb{E}\left[(\bar{a}^{T_b})^2\right]}}{K^2\alpha^3}\sqrt{\left[\log M + 2\log\left(M/\alpha\right)\mathbb{E}\left[\text{TV}\left(p_\star^{T_b:(T_{b+1}-1)}\right)\right]\right]}(T_{b+1} - T_b)$$

$$+ \frac{M}{\alpha}\sqrt{\frac{(T_{b+1} - T_b)\mathbb{E}\left[\bar{a}^{T_b}\right]}{8K}}(C + 2\log E_b)$$

$$\leq \frac{(2C+1)\sqrt{2}M^3\sqrt{\mathbb{E}\left[(\bar{a}^1)^2\right]}}{K^2\alpha^3}\sqrt{\left[\log M + 2\log\left(M/\alpha\right)\text{TV}\left(p_\star^{1:T}\right)\right]}(\sqrt{2})^{b-1}$$

$$+ \frac{M}{\alpha}\sqrt{\frac{\mathbb{E}\left[\bar{a}^1\right]}{8K}}(C + 2\log E)(\sqrt{2})^{b-1},$$

where $E$ is defined in equation 14. Since $B = \lfloor\log_2(T+1)\rfloor$, we have $T_B \leq T \leq T_{B+1} - 1$, which implies that $1 \leq T - T_B + 1 \leq T_{B+1} - T_B = 2^B$. Thus, we can similarly obtain

$$\mathbb{E}\left[\sum_{t=T_B}^{T} l_t(\hat{p}^t) - \sum_{t=T_B}^{T} l_t(p_\star^t)\right]$$

$$\leq \frac{(2C+1)\sqrt{2}M^3\sqrt{\mathbb{E}\left[(\bar{a}^1)^2\right]}}{K^2\alpha^3}\sqrt{\left[\log M + 2\log\left(M/\alpha\right)\text{TV}\left(p_\star^{1:T}\right)\right]}(\sqrt{2})^B$$

$$+ \frac{M}{\alpha}\sqrt{\frac{\mathbb{E}\left[\bar{a}^1\right]}{8K}}(C + 2\log E)(\sqrt{2})^B.$$

The result follows combing the last two displays.

### D.6 PROOF OF THEOREM 2

Our proof follows the similar technique used in the proof of Theorem 2.1 of Ghadimi & Lan (2013). Our key novel technique is the construction of a ghost subset that is drawn from $[M]$ from the comparator sampling distribution. The ghost subset is only constructed for theoretical purpose and does not need to be computed in practice. We only show the proof of $R_1$, $R_0$ and $R_2$ can be then derived in a similar fashion.

Let $\delta^t = g^t - \nabla F(w^t)$. Under Assumption 3, by (1.6) of Ghadimi & Lan (2013), we have

$$\begin{aligned}
F\left(w^{t+1}\right) &\leq F\left(w^t\right) + \left\langle\nabla F\left(w^t\right), w^{t+1} - w^t\right\rangle + \frac{L}{2}\mu^2\left\|g^t\right\|^2 \\
&= F\left(w^t\right) - \mu\left\langle\nabla F\left(w^t\right), g^t\right\rangle + \frac{L}{2}\mu^2\left\|g^t\right\|^2 \\
&= F\left(w^t\right) - \mu\left\|\nabla F\left(w^t\right)\right\|^2 - \mu\left\langle\nabla F\left(w^t\right), \delta^t\right\rangle \\
&\quad + \frac{L}{2}\mu^2\left[\left\|\nabla F\left(w^t\right)\right\|^2 + 2\left\langle\nabla F\left(w^t\right), \delta^t\right\rangle + \left\|\delta^t\right\|^2\right] \\
&= F\left(w^t\right) - \left(\mu - \frac{L}{2}\mu^2\right)\left\|\nabla F\left(w^t\right)\right\|^2 - \left(\mu - L\mu^2\right)\left\langle\nabla F\left(w^t\right), \delta^t\right\rangle + \frac{L}{2}\mu^2\left\|\delta^t\right\|^2.
\end{aligned}$$
(30)

Note that $\mathbb{E}\left[g^t \mid w^t, \hat{p}^t\right] = \nabla F(w^t)$, thus we have $\mathbb{E}\left[\delta^t \mid w^t, \hat{p}^t\right] = 0$, and

$$\mathbb{E}\left[\left\langle\nabla F\left(w^t\right), \delta^t\right\rangle\right] = \mathbb{E}\left[\mathbb{E}\left[\left\langle\nabla F\left(w^t\right), \delta^t\right\rangle \mid w^t, \hat{p}^t\right]\right] = 0.$$
(31)

On the other hand, we can assume that there is a ghost subset of clients $\tilde{S}^t$ with $|\tilde{S}^t| = K$, which is drawn from $[M]$ with sampling distribution $q_\star$. Besides, we let

$$\tilde{g}^t := \frac{1}{MK} \sum_{m \in \tilde{S}^t} \frac{g_m^t}{q_m^\star}.$$

Then we have

$$\mathbb{E}_{S^t} \left[ \left\| \delta^t \right\|^2 \mid w^t, \hat{p}^t \right] = \mathbb{E}_{S^t} \left[ \left\| g^t - J^t + J^t - \nabla F(w^t) \right\|^2 \mid w^t, \hat{p}^t \right]$$

$$= \mathbb{E}_{S^t} \left[ \left\| g^t - J^t \right\|^2 \mid w^t, \hat{p}^t \right] + \left\| J^t - \nabla F(w^t) \right\|^2$$

$$= l_t \left( \hat{p}^t \right) - \left\| J^t \right\|^2 + \left\| J^t - \nabla F(w^t) \right\|^2$$

$$= l_t \left( q_\star \right) - \left\| J^t \right\|^2 + \left\| J^t - \nabla F(w^t) \right\|^2 + l_t \left( \hat{p}^t \right) - l_t \left( q_\star \right)$$

$$= \mathbb{E}_{\tilde{S}^t} \left[ \left\| \tilde{g}^t - \nabla F(w^t) \right\|^2 \mid w^t, \hat{p}^t \right] + l_t \left( \hat{p}^t \right) - l_t \left( q_\star \right).$$

Since

$$\mathbb{E} \left[ \left\| \tilde{g}^t - \nabla F(w^t) \right\|^2 \mid w^t, \hat{p}^t \right]$$

$$= \mathbb{E} \left[ \left\| \frac{1}{MK} \sum_{m \in \tilde{S}^t} \frac{g_m^t}{q_m^\star} - \nabla F(w^t) \right\|^2 \mid w^t, \hat{p}^t \right]$$

$$\leq 2\mathbb{E} \left[ \left\| \frac{1}{MK} \sum_{m \in \tilde{S}^t} \frac{g_m^t}{q_m^\star} - \frac{1}{MK} \sum_{m \in \tilde{S}^t} \frac{\nabla F_m(w^t)}{q_m^\star} \right\|^2 \mid w^t, \hat{p}^t \right]$$

$$+ 2\mathbb{E} \left[ \left\| \frac{1}{MK} \sum_{m \in \tilde{S}^t} \frac{\nabla F_m(w^t)}{q_m^\star} - \nabla F(w^t) \right\|^2 \mid w^t, \hat{p}^t \right]$$

$$= \frac{2}{M^2 K^2} \mathbb{E} \left[ \sum_{m \in \tilde{S}^t} \mathbb{E} \left[ \frac{\left\| g_m^t - \nabla F_m(w^t) \right\|^2}{(q_m^\star)^2} \right] \mid w^t, \hat{p}^t \right]$$

$$+ \frac{2}{K} \left( \frac{1}{M^2} \sum_{m=1}^{M} \frac{\left\| \nabla F_m(w^t) \right\|^2}{q_m^\star} - \left\| \nabla F_m(w^t) \right\|^2 \right)$$

$$\leq \frac{2\sigma^2}{M^2 KB} \sum_{m=1}^{M} \frac{1}{q_m^\star} + \frac{2\zeta_1^2}{K} \leq \frac{2\sigma^2}{KB\alpha} + \frac{2\zeta_1^2}{K},$$

where the penultimate line follows that $\mathbb{E} \left[ \left\| g_m^t - \nabla F_m(w^t) \right\|^2 \right] \leq \sigma^2/B$ and the definition of $\zeta_1$, and the last line follows that $q_m^\star \geq \alpha/M$.

Thus, we have

$$\mathbb{E} \left[ \left\| \delta^t \right\|^2 \mid w^t, \hat{p}^t \right] \leq \frac{2\sigma^2}{KB\alpha} + \frac{2\zeta_1^2}{K} + l_t \left( \hat{p}^t \right) - l_t \left( q_\star \right),$$

which implies that

$$\sum_{t=0}^{T-1} \mathbb{E} \left[ \left\| \delta^t \right\|^2 \right] \leq \frac{2T\sigma^2}{KB\alpha} + \frac{2T\zeta_1^2}{K} + \mathbb{E} \left[ \sum_{t=0}^{T-1} l_t \left( \hat{p}^t \right) - \sum_{t=0}^{T-1} l_t \left( q_\star \right) \right]$$

$$= \frac{2T\sigma^2}{KB\alpha} + \frac{2T\zeta_1^2}{K} + \text{D-Regret}_T(q^\star). \tag{32}$$

Combine equation 30, equation 31 and equation 32, we have

$$\left(\mu - \frac{L}{2}\mu^2\right) \sum_{t=0}^{T-1} \mathbb{E}\left[\left\|\nabla F\left(w^t\right)\right\|^2\right]$$

$$\leq F(w^1) - F(w^T) + \frac{L}{2}\mu^2 \left(\frac{2T\sigma^2}{KB\alpha} + \frac{2T\zeta_1^2}{K} + \text{D-Regret}_T(q^\star)\right)$$

$$\leq D^F + \frac{L}{2}\mu^2 \left(\frac{2T\sigma^2}{KB\alpha} + \frac{2T\zeta_1^2}{K} + \text{D-Regret}_T(q^\star)\right).$$

Since $\mu \leq 1/L$, thus $(\mu - \frac{L}{2}\mu^2) = \mu(1 - \frac{L}{2}\mu) \geq \mu/2$, thus

$$\frac{1}{T}\sum_{t=0}^{T-1} \mathbb{E}\left[\left\|\nabla F\left(w^t\right)\right\|^2\right] \leq \frac{2D^F}{T\mu} + L\mu \left(\frac{2\sigma^2}{KB\alpha} + \frac{2\zeta_1^2}{K} + \frac{\text{D-Regret}_T(q^\star)}{T}\right).$$

Finally, let

$$\mu = \min\left\{1/L, (1/\sigma)\sqrt{D^F KB\alpha/(LT)}, (1/\zeta_1)\sqrt{D^F K/(TL)}, \sqrt{D^F/(L\text{D-Regret}_T(q^\star))}\right\}, \tag{33}$$

then we have

$$\frac{1}{T}\sum_{t=0}^{T-1} \mathbb{E}\left[\left\|\nabla F\left(w^t\right)\right\|^2\right]$$

$$\lesssim \frac{D^F}{T}\max\left\{L, \sigma\sqrt{\frac{LT}{D^F KB\alpha}}, \zeta_1\sqrt{\frac{TL}{D^F K}}, \sqrt{\frac{L\text{D-Regret}_T(q^\star))}{D^F}}\right\}$$

$$+ \frac{\sigma\sqrt{D^F L}}{\sqrt{TKB\alpha}} + \frac{\zeta_1\sqrt{D^F L}}{\sqrt{TK}} + \frac{\sqrt{D^F L}\sqrt{\text{D-Regret}_T(q^\star))}}{T}$$

$$\lesssim \frac{D^F}{T}\left(L + \sigma\sqrt{\frac{LT}{D^F KB\alpha}} + \zeta_1\sqrt{\frac{TL}{D^F K}} + \sqrt{\frac{L\text{D-Regret}_T(q^\star))}{D^F}}\right)$$

$$+ \frac{\sigma\sqrt{D^F L}}{\sqrt{TKB\alpha}} + \frac{\zeta_1\sqrt{D^F L}}{\sqrt{TK}} + \frac{\sqrt{D^F L}\sqrt{\text{D-Regret}_T(q^\star))}}{T}$$

$$\lesssim \frac{D^F L}{T} + \frac{\sigma\sqrt{D^F L}}{\sqrt{TKB\alpha}} + \frac{\zeta_1\sqrt{D^F L}}{\sqrt{TK}} + \frac{\sqrt{D^F L}\sqrt{\text{D-Regret}_T(q^\star))}}{T}.$$

## E USEFUL LEMMAS

**Lemma 1.** *Suppose that $f$ is a differentiable convex function defined on $\text{dom} f$, and $\mathcal{X} \subseteq \text{dom} f$ is a closed convex set. Then $x$ is the minimizer of $f$ on $\mathcal{X}$ if and only if*

$$\nabla f(x)^\top (y - x) \geq 0 \quad \text{for all } y \in \mathcal{X}.$$

*Proof.* See Section 4.2.3 of Boyd et al. (2004). $\square$

**Lemma 2.** *For $q \in \mathcal{P}_{M-1}$ we have $D_\Phi(q \,\|\, p^{\text{unif}}) \leq \log M$, where $\Phi$ is the unnormalized negative entropy.*

*Proof.* Since $\Phi(q) = \sum_{m=1}^M q_m(\log q_m - 1) \leq 0$, $\Phi(p^{\text{unif}}) = -\log M$, and

$$\langle \nabla\Phi(p^{\text{unif}}), q - p^{\text{unif}}\rangle = \sum_{m=1}^M (q_m - \frac{1}{M})\log\frac{1}{M} = 0,$$

we have $D_\Phi(q \,\|\, p) \leq \log M$. $\square$

**Lemma 3** (Hoeffding's Inequality). *Let $X$ be a random variable with $a \leq X \leq b$ for $a, b \in \mathbb{R}$. Then for all $s \in \mathbb{R}$, we have*

$$\log \mathbb{E}\left[e^{sX}\right] \leq s\mathbb{E}[X] + \frac{s^2(b-a)^2}{8}.$$

*Proof.* See Section 2 of Wainwright (2019). $\square$

**Lemma 4** (Based on Exercise 26.12 of Lattimore & Szepesvári (2020)). *Let $\alpha \in [0, 1]$, $\mathcal{A} = \mathcal{P}_{M-1} \cap [\alpha/M, 1]^M$, $\mathcal{D} = [0, \infty)^M$, and $\Phi = \Phi_e$ is the unnormalised entropy on $\mathcal{D}$. For $y \in [0, \infty)^M$, let $x = \arg\min_{v \in \mathcal{A}} D_\Phi(v\|y)$. Suppose $y_1 \leq y_2 \leq \cdots \leq y_M$. Let $m^\star$ be the smallest value such that*

$$y_{m^\star}\left(1 - \frac{m^\star - 1}{M}\alpha\right) > \frac{\alpha}{M}\sum_{m=m^\star}^{M} y_m.$$

*Then*

$$x_m = \begin{cases} \frac{\alpha}{M} & \text{if } m < m^\star \\ \frac{(1 - \frac{m^\star - 1}{M}\alpha)y_m}{\sum_{n=m^\star}^{M} y_n} & \text{otherwise.} \end{cases}$$

*Proof.* Consider the following constrained optimization problem:

$$\min_{u \in [0,\infty)^M} \sum_{m=1}^{M} u_m \log \frac{u_m}{y_m},$$

$$\text{s.t. } \sum_{m=1}^{M} u_m = 1,$$

$$u_m \geq \frac{\alpha}{M}, \qquad m \in [M].$$

Since $x$ is the solution to this problem, by the optimality condition, there exists $\lambda, \nu_1, \ldots, \nu_M \in \mathbb{R}$ such that

$$\log \frac{x_m}{y_m} + 1 - \lambda - \nu_m = 0, \qquad m \in [M], \tag{34}$$

$$\sum_{m=1}^{M} x_m = 1, \tag{35}$$

$$x_m - \frac{\alpha}{M} \geq 0, \qquad m \in [M], \tag{36}$$

$$\nu_m \geq 0, \qquad m \in [M], \tag{37}$$

$$\nu_m\left(x_m - \frac{\alpha}{M}\right) = 0, \qquad m \in [M]. \tag{38}$$

By equation 34, we have $x_m = y_m \exp(-1 + \lambda + \nu_m)$. By equation 37 and equation 38, when $x_m = \alpha/M$, we have $x_m = y_m \exp(-1 + \lambda + \nu_m) \geq y_m \exp(-1 + \lambda)$; when $x_m > \alpha/M$, we have $x_m = y_m \exp(-1 + \lambda)$. Assume that $x_1 = \cdots = x_{m^\star - 1} = \alpha/M < x_{m^\star} \leq \cdots \leq x_M$. Then

$$1 = \sum_{m=1}^{M} x_m = (m^\star - 1)\frac{\alpha}{M} + \exp(-1 + \lambda) \cdot \sum_{m=m^\star}^{M} y_m,$$

which implies that

$$\exp(-1 + \lambda) = \frac{1 - (m^\star - 1)\frac{\alpha}{M}}{\sum_{m=m^\star}^{M} y_m}. \tag{39}$$

Thus, we have

$$x_{m^\star} = y_{m^\star}\exp(-1 + \lambda) = y_{m^\star}\frac{1 - (m^\star - 1)\frac{\alpha}{M}}{\sum_{m=m^\star}^{M} y_m} > \frac{\alpha}{M},$$

which implies that

$$y_{m^\star} \left(1 - \frac{m^\star - 1}{M}\alpha\right) > \frac{\alpha}{M} \sum_{m=m^\star}^{M} y_m. \tag{40}$$

To complete the proof, we then only need to show that

$$y_{m'} \left(1 - \frac{m' - 1}{M}\alpha\right) \leq \frac{\alpha}{M} \sum_{m=m'}^{M} y_m \tag{41}$$

for all $1 \leq m' \leq m^\star - 1$. The result then follows from equation 40 and equation 41. To prove equation 41, recall that for any $1 \leq m' \leq m^\star - 1$, we have $\alpha/M = y_{m'} \exp(-1 + \lambda + \nu_{m'})$, and because $y_1 \leq \cdots \leq y_M$, we have $\nu_1 \geq \cdots \geq \nu_{m^\star - 1}$. This way, we have

$$(m^\star - m') \frac{\alpha}{M} = \sum_{m=m'}^{m^\star - 1} y_m \exp\left(-1 + \lambda + \nu_m\right) \leq \exp\left(-1 + \lambda + \nu_{m'}\right) \sum_{m=m'}^{m^\star - 1} y_m. \tag{42}$$

On the other hand, by equation 39, we have

$$1 - (m^\star - 1)\frac{\alpha}{M} = \exp(-1 + \lambda) \sum_{m=m^\star}^{M} y_m \leq \exp\left(-1 + \lambda + \nu_{m'}\right) \sum_{m=m^\star}^{M} y_m. \tag{43}$$

Combining equation 42 and equation 43, we have

$$
\begin{aligned}
\frac{1 - (m' - 1)\frac{\alpha}{M}}{\sum_{m=m'}^{M} y_m} &= \frac{1 - (m^\star - 1)\frac{\alpha}{M} + (m^\star - m')\frac{\alpha}{M}}{\sum_{m=m'}^{M} y_m} \\
&\leq \frac{\exp\left(-1 + \lambda + \nu_{m'}\right) \left(\sum_{m=m'}^{m^\star - 1} y_m + \sum_{m=m^\star}^{M} y_m\right)}{\sum_{m=m'}^{M} y_m} \\
&= \exp\left(-1 + \lambda + \nu_{m'}\right) \\
&= \frac{\frac{\alpha}{M}}{y_{m'}},
\end{aligned}
$$

which then implies equation 41. $\qquad\square$

## F  SYNTHETIC EXPERIMENTS

In this section, we use synthetic data to demonstrate the performance of Adaptive-OSMD Sampler. We compare our method against uniform sampling in Section F.1 and compare against other bandit feedback online learning samplers in Section F.2. In addition, we examine the robustness of Adaptive-OSMD Sampler to the choice of $\alpha$ in Section F.3, while in Section F.4, we compare it with the sampling without replacement variant discussed in Section C.5. Finally, in Section F.6, we show some empirical observation for regret analysis.

We generate data as follows. We set the number of clients as $M = 100$, and each client has $n_m = 100$ samples, $m \in [M]$. Samples on each client are generated as

$$y_{m,i} = \langle w_\star, x_{m,i} \rangle + N(0, 0.1^2), \qquad i \in [n_m], \tag{44}$$

where the coefficient vector $w_\star \in \mathbb{R}^d$ has elements generated as i.i.d. $N(10, 3)$, and the feature vector $x_{m,i} \in \mathbb{R}^d$ is generated as $x_{m,i} \sim N(0, \Sigma_m)$, where $\Sigma_m = s_m \cdot \Sigma$, $\Sigma$ is a diagonal matrix with $\Sigma_{jj} = \kappa^{(j-1)/(d-1)-1}$ and $\kappa > 0$ is the condition number of $\Sigma$. We generate $\{s_m\}_{m=1}^{M}$ i.i.d. from $e^{N(0,\sigma^2)}$ and rescale them as $s_m \leftarrow (s_m / \max_{m \in [M]} s_m) \times 10$ so that $s_m \leq 10$ for all $m \in [M]$. In this setting, $\kappa$ controls the difficulty of each problem when solved separately, while $\sigma$ controls the level of heterogeneity across clients. In all experiments, we fix $\kappa = 25$, which corresponds to a hard problem, and change $\sigma$ to simulate different heterogeneity levels. We expect that uniform sampling suffers when the heterogeneity level is high. The dimension $d$ of the problem is set as $d = 10$. The results are averaged over 10 independent runs.

We use the mean squared error loss defined as

$$L(w) = \frac{1}{M} \sum_{m=1}^{M} L_m(w), \qquad \text{where} \qquad L_m(w) = \frac{1}{2n_m}(y_{m,i} - \langle w_\star, x_{m,i}\rangle)^2.$$

We use the stochastic gradient descent to make global updates. At each round $t$, we choose a subset of $K = 5$ clients, denoted as $S^t$. For each client $m \in S^t$, we choose a mini-batch of samples, $\mathcal{B}_m^t$, of size $\bar{B} = 10$, and compute the mini-batch stochastic gradient. The parameter $w$ is updated as

$$w^{t+1} = w^t + \frac{\mu_{\text{SGD}}}{MK\bar{B}} \sum_{m \in S^t} \frac{1}{p_m^t} \sum_{i \in \mathcal{B}_m^t} (y_{m,i} - \langle w_\star, x_{m,i}\rangle) \cdot x_{m,i},$$

where $\mu_{\text{SGD}}$ is the learning rate, set as $\mu_{\text{SGD}} = 0.1$ in simulations.

In all experiments, we set $\alpha$ in Adaptive-OSMD Sampler as $\alpha = 0.4$. The tuning parameters for MABS, VRB and Avare are set as in their original papers.

**Computational resources and amount of compute.** All the computation was done on a personal laptop. The synthetic data experiments are computed by CPU (Intel(R) Core(TM) i7-9750H CPU @ 2.60GHz 2.59 GHz). Each run of all experiments in this section took less than 10 minutes.

## F.1 ADAPTIVE-OSMD SAMPLER VS UNIFORM SAMPLING

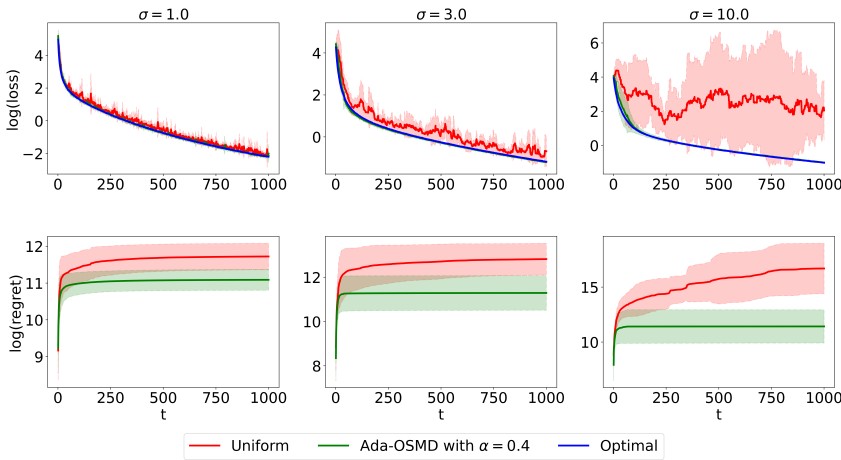

Figure 1: The training loss (top row) and the cumulative regret (bottom row) of Adaptive-OSMD Sampler vs Uniform vs Optimal with $\sigma = 1.0$, $\sigma = 3.0$, and $\sigma = 10.0$. The solid line denotes the mean and the shaded region covers mean $\pm$ standard deviation across independent runs.

The results of the training process and the cumulative regret are shown in Figure 1. For the training loss, we see that when the heterogeneity level is low ($\sigma = 1.0$), the uniform sampling performs as well as Adaptive-OSMD Sampler and theoretically optimal sampling; however, as the heterogeneity level increases, the performance of uniform sampling gradually suffers; when $\sigma = 10.0$, uniform sampling performs poorly. On the other hand, Adaptive-OSMD Sampler performs well across all levels of heterogeneity and is very close to the theoretically optimal sampling. Similarly, for the cumulative regret, when the heterogeneity level is low, the cumulative regret of uniform sampling is close to Adaptive-OSMD Sampler; however, when the heterogeneity level increases, the cumulative regret of uniform sampling gets much larger than Adaptive-OSMD Sampler. Based on the above results, we can conclude that while the widely used choice of uniform sampling may be reasonable when heterogeneity is low, our proposed sampling strategy is robust across different levels of heterogeneity, and thus should be considered as the default option.

## F.2 ADAPTIVE-OSMD SAMPLER VS MABS VS VRB VS AVARE

We compare Adaptive-OSMD Sampler to other bandit feedback online learning samplers: MABS (Salehi et al., 2017), VRB (Borsos et al., 2018) and Avare (Hanchi & Stephens, 2020).

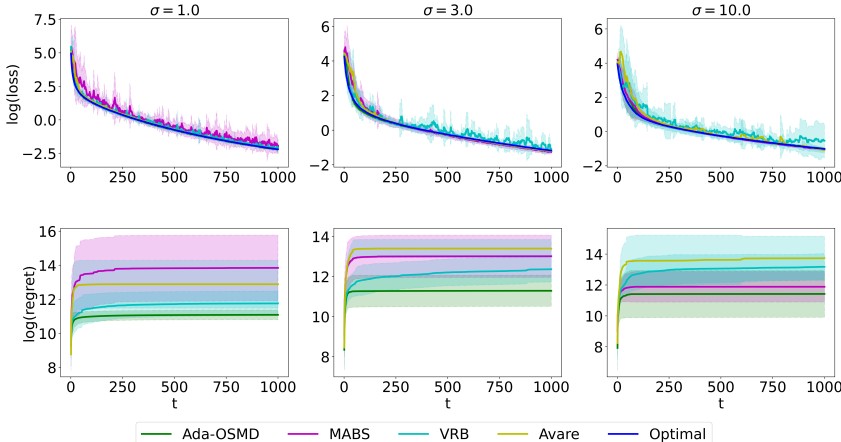

Figure 2: The training loss (top row) and the cumulative regret (bottom row) of Adaptive-OSMD Sampler vs MABS vs VRB vs Avare with $\sigma = 1.0$, $\sigma = 3.0$ and $\sigma = 10.0$. The solid line denotes the mean and the shaded region covers mean $\pm$ standard deviation across independent runs.

Training loss and cumulative regret are shown in Figure 2. We see that while VRB and Avare perform better when the heterogeneity level is low and MABS performs better when the heterogeneity level is high, Adaptive-OSMD Sampler always achieves the best in both training loss and cumulative regret across all different levels of heterogeneity. Thus, we conclude that Adaptive-OSMD is a better choice than other online learning samplers.

### F.3    ROBUSTNESS OF ADAPTIVE-OSMD SAMPLER TO THE CHOICE OF $\alpha$

We examine the robustness of Adaptive-OSMD Sampler to the choice of $\alpha$. We run Adaptive-OSMD Sampler separately for each $\alpha \in \{0.01, 0.1, 0.4, 0.7, 0.9, 1.0\}$. Note that when $\alpha = 1.0$, the Adaptive-OSMD Sampler outputs a uniform distribution. Training loss and cumulative regret are shown in Figure 3. We observe that Adaptive-OSMD Sampler is robust to the choice of $\alpha$, and performs well as long as $\alpha$ is not too close to zero or too close to one.

### F.4    EXPERIMENTS ON SAMPLING WITH REPLACEMENT VS WITHOUT REPLACEMENT

We compare sampling with replacement and sampling without replacement when used together with Adaptive-OSMD sampler. Sampling without replacement is described in Section C.5. Training loss and cumulative regret are shown in Figure 4. We observe that using sampling with replacement results in a slightly smaller cumulative regret and a slightly better training loss. However, these differences are not significant.

### F.5    DYNAMIC SAMPLING DISTRIBUTION V.S. FIXED SAMPLING DISTRIBUTION

In this paper, we allow both our sampling distribution and competitor sampling distribution to change over time, while previous studies either use fixed sampling distribution (Zhao & Zhang, 2015; Needell et al., 2016) or they compare against fixed sampling distribution (Namkoong et al., 2017; Salehi et al., 2017; Borsos et al., 2018; 2019). In this section, we show that under certain settings, dynamic sampling distribution can achieve significant advantage over fixed sampling distribution. More specifically, we compare Adaptive-OSMD Sampler with the Lipschitz constant based importance sampling distribution proposed by Zhao & Zhang (2015); Needell et al. (2016), which we denote as $p^{\text{IS}}$.

We still use the same model as in equation 44 to generate data. but we generate $w_\star$ and $x_{m,i}$ differently. Motivated by Zhao et al. (2022), for each $m \in [M]$, we choose uniformly at random one dimension among $\mathbb{R}^d$, denoted as $\text{supp}(m) \in [d]$, as the support of $x_{m,i}$ for all $i \in [n_m]$, while the remaining dimensions of $x_{m,i}$ are set to be zero. The nonzero dimension of $x_{m,i}$ is generated

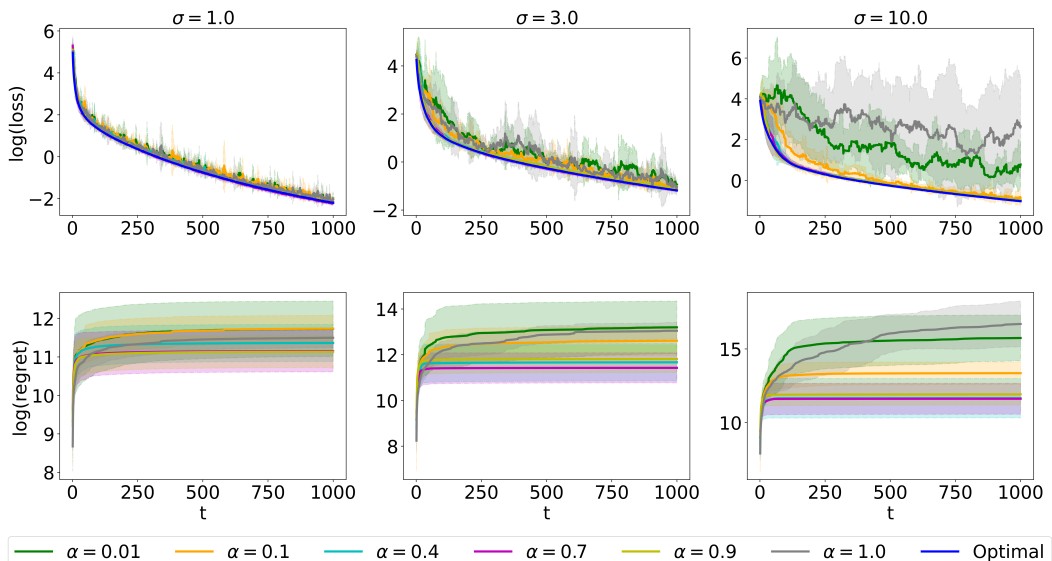

Figure 3: The training loss (top row) and the cumulative regret (bottom row) of Adaptive-OSMD Sampler with different choices of $\alpha$ under $\sigma = 1.0$, $\sigma = 3.0$ and $\sigma = 10.0$. The solid line denotes the mean and the shaded region covers mean $\pm$ standard deviation across independent runs.

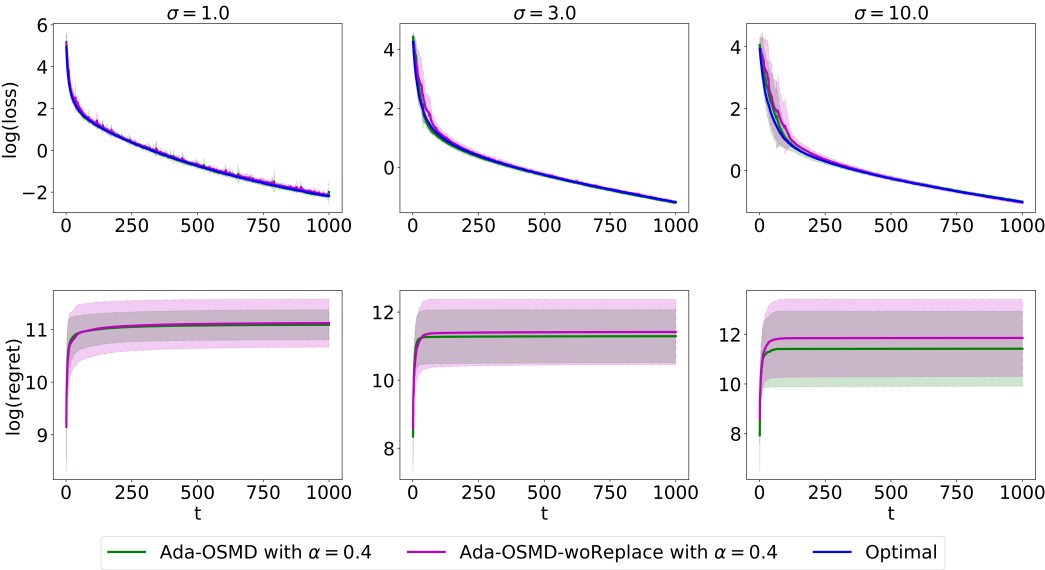

Figure 4: The training loss (top row) and the cumulative regret (bottom row) of Adaptive-OSMD Sampler with replacement vs without replacement across $\sigma = 1.0$, $\sigma = 3.0$ and $\sigma = 10.0$. The solid line denotes the mean and the shaded region covers mean $\pm$ standard deviation across independent runs.

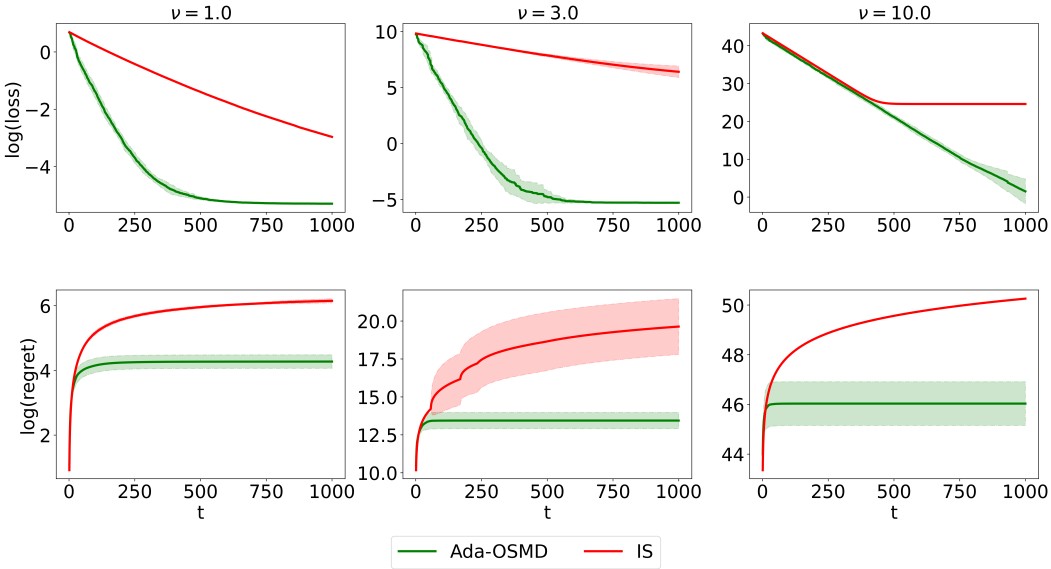

Figure 5: The training loss (top row) and the cumulative regret (bottom row) of Adaptive-OSMD Sampler vs $p^{\text{IS}}$ across $\nu = 1.0$, $\nu = 3.0$ and $\nu = 10.0$. The solid line denotes the mean and the shaded region covers mean $\pm$ standard deviation across independent runs.

from $N(1.0, 0.1^2)$. The entries of $w_\star$ are generated i.i.d. from $e^{N(0,\nu^2)}$. Therefore, $\nu$ controls the variance of entries of $w_\star$.

Besides, we choose the optimal stepsize from the set $\{1.0, 0.5, 0.1, 0.05, 0.01\}$ for each method separately. The final result is shown in Figure 5. We see that Adaptive-OSMD Sampler performs better than $p^{\text{IS}}$ across all levels of $\nu$. Note that in practice, in order to implement $p^{\text{IS}}$, we need prior information about Lipschitz constants of $L_m(\cdot)$'s, while Adaptive-OSMD Sampler does not need prior information. This way, our proposed method does not only have better practical performance, but also requires less prior information.

### F.6 EMPIRICAL OBSERVATION OF REGRET

In Corollary 1, we see that the upper bound for the regret of OSMD sampler with respect to any comparator sequence $q^{1:T}$ depends on two important quantities: $\sum_{t=1}^{T}(\bar{a}^t)^2$ and $\text{TV}(q^{1:T})$. In this section, we first empirically show how $\bar{a}^t$ and $\sum_{l=1}^{t}(\bar{a}^l)^2$ grow with $t$ in practice. Then, we use $p_\star^{1:T}$ as the comparator and show how $\text{TV}(p_\star^{1:t})$ grow with $t$ empirically. Finally, we also empirically show how the regret D-Regret$_T(p_\star^{1:T})$ grows. The experimental setting of this section is the same as Section F.

In Figure 6, we plot both $\bar{a}^t$ and $\sum_{l=1}^{t}(\bar{a}^l)^2$ over $t$. We see that $\bar{a}^t$ drops to zero fastly. As a result $\sum_{l=1}^{t}(\bar{a}^l)^2$ converges to a constant as $t \to \infty$. This way, we empirically have $\sum_{l=1}^{t}(\bar{a}^l)^2 = O(1)$. In Figure 7, we show how $\text{TV}(p_\star^{1:t})$ grows with $t$. We see that $\text{TV}(p_\star^{1:t}) = O(t)$ empirically, which is consistent with the worst-case upper bound. Based on this result, the best upper bound of D-Regret$_T(p_\star^{1:T})$ we can hope in practice is $O(\sqrt{T})$. However, as shown in Figure 8, in practice, we have D-Regret$_T(p_\star^{1:T})$ converge to constant. This observation indicates that the upper bound we obtained might still be loose compared to practical result.

## G REAL DATA EXPERIMENT

We compare Adaptive-OSMD Sampler with uniform sampling and other online learning samplers including MABS (Salehi et al., 2017), VRB (Borsos et al., 2018) and Avare (Hanchi & Stephens, 2020) on real data. We use three commonly used computer vision data sets: MNIST (LeCun &

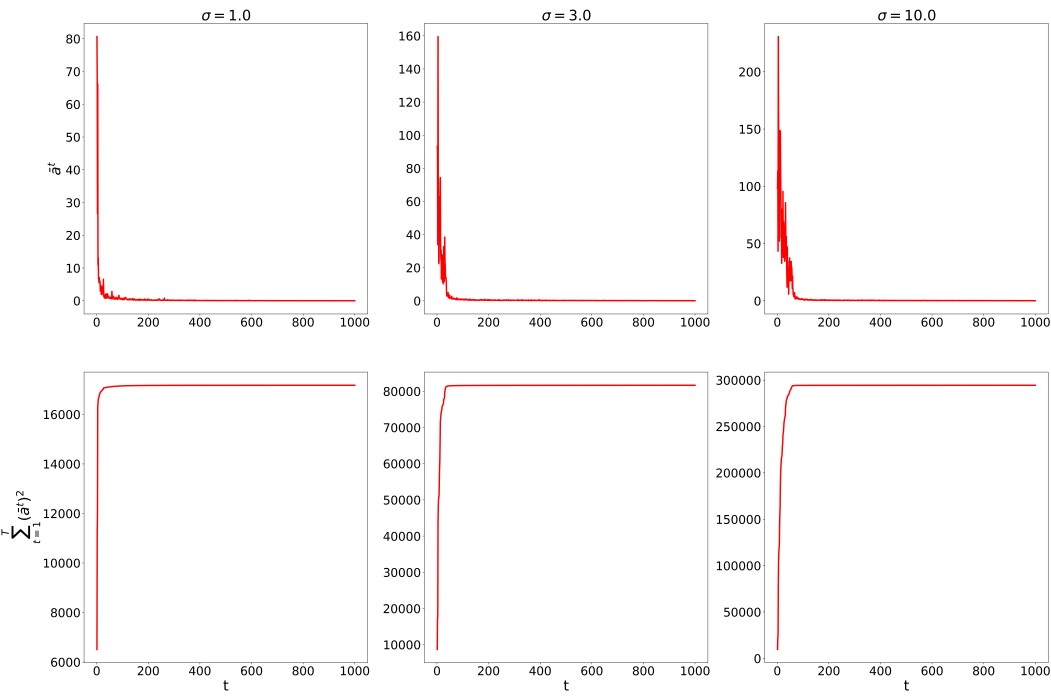

Figure 6: Plots of how $\bar{a}^t$ and $\sum_{l=1}^{t}(\bar{a}^l)^2$ grow with $t$.

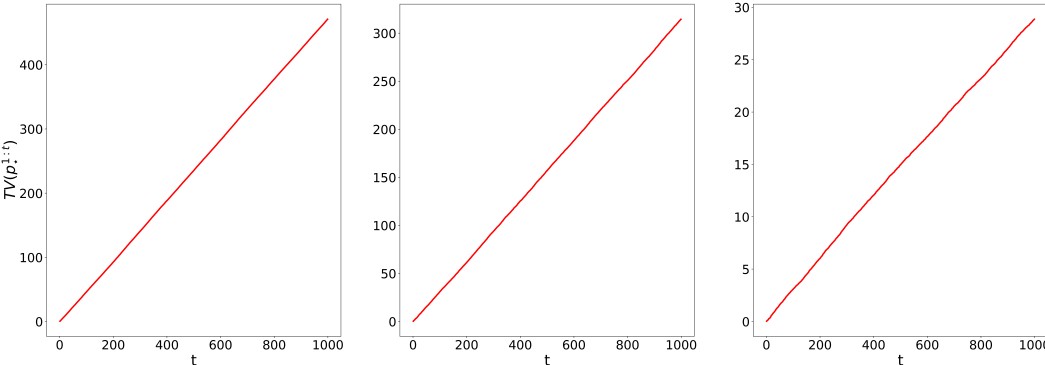

Figure 7: Plots of how $\mathrm{TV}(p_\star^{1:t})$ grows with $t$.

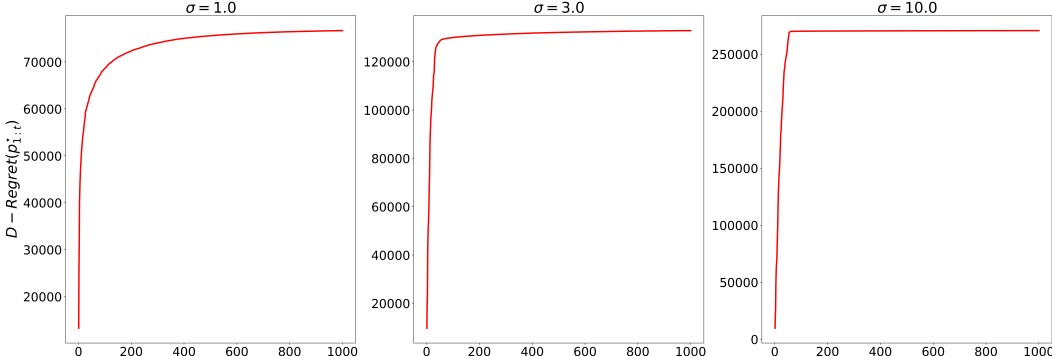

Figure 8: Plots of how D-Regret$_T(p_\star^{1:T})$ grows with $t$.

Cortes, 2010)[8], KMINIST (Clanuwat et al., 2018)[9], and FMINST (Xiao et al., 2017)[10]. We set the number of devices to be $M = 500$. To better simulate the situation where our method brings significant convergence speed improvement, we create a highly skewed sample size distribution of the training set among clients: 65% of clients have only one training sample, 20% of clients have 5 training samples, 10% of clients have 30 training samples, and 5% of clients have 100 training samples. This setting tries to illustrate a real-life situation where most of the data come from a small fraction of users, while most of the users have only a small number of samples. The skewed sample size distribution is common in other FL data sets, such as LEAF (Caldas et al., 2018). The sample size distribution in the training set is shown in Figure 9. In addition, each client has 10 validation samples used to measure the prediction accuracy of the model over the training process.

We use a multi-class logistic regression model. For a given gray scale picture with the label $y \in \{1, 2, \ldots, C\}$, we unroll its pixel matrix into a vector $x \in \mathbb{R}^p$. Given a parameter matrix $W \in \mathbb{R}^{C \times P}$, the training loss function defined in equation 1 is

$$\phi(W; x, y) := l_{\text{CE}} \left( \varsigma(Wx) ; y \right),$$

where $\varsigma(\cdot) : \mathbb{R}^C \to \mathbb{R}^C$ is the softmax function defined as

$$[\varsigma(x)]_i = \frac{\exp(x_i)}{\sum_{j=1}^{K} \exp(x_j)}, \quad \text{for all } x \in \mathbb{R}^C,$$

and $l_{\text{CE}}(x ; y) = \sum_{i=1}^{C} \mathbb{1}(y = i) \log x_i$, $x \in \mathbb{R}^C$, $y \in \{1, \ldots, C\}$, is the cross-entropy function.

We use the same algorithms and tuning parameters as in Section F. Learning rate in SGD is set to $0.075$ for MNIST and KMNIST, and is set to $0.03$ for FMNIST. The total number of communication rounds is to $1,000$. In each round of communication, we choose $K = 10$ clients to participate (2% of total number of clients). For a chosen client $m$, we compute its local mini-batch gradient with the batch size equal to $\min\{5, n_m\}$, where $n_m$ is the training sample size on the client $m$.

Figure 10 shows both the training loss and validation accuracy. Each figure shows the average performance over 5 independent runs. We use the same random seed for both Adaptive-OSMD Sampler and competitors, and change random seeds across different runs. The main focus is on minimizing the training loss, and the validation accuracy is only included for completeness. We observe that Adaptive-OSMD Sampler performs better than uniform sampling and other online learning samplers across all data sets. Given the cheap computational cost and the significant practical advantage, we recommend using Adaptive-OSMD Sampler as the default option in practice. Besides, for the completeness of the paper, we also include the results under homogeneous setting where the sample sizes are balanced across different clients. The result is shown in Figure 11, where we see that all methods perform similarly.

**Computational resources and amount of compute.** All the computation was done on a personal laptop. The real data experiments are computed by GPU (NVIDIA GeForce RTX 2070 with Max-Q Design). Each run of all experiments in this section took less than 15 minutes.

## H  DISCUSSION

We studied the client sampling problem in FL. We proposed an online learning with bandit feedback approach to tackle client sampling. We used online stochastic mirror descent to solve the online learning problem and applied the online ensemble method with doubling trick to choose the tuning parameters. We established an upper bound on the dynamic regret relative to the theoretically optimal sequence of sampling distributions. The total variation of the comparator is explicitly included in the upper bound as a measure of the difficulty of the problem. Extensive numerical experiments

---

[8]Yann LeCun and Corinna Cortes hold the copyright of MNIST dataset, which is a derivative work from original NIST datasets. MNIST dataset is made available under the terms of the Creative Commons Attribution-Share Alike 3.0 license.

[9]KMNIST dataset is licensed under a permissive CC BY-SA 4.0 license, except where specified within some benchmark scripts.

[10]FMNIST dataset is under The MIT License (MIT) Copyright © [2017] Zalando SE, https://tech.zalando.com

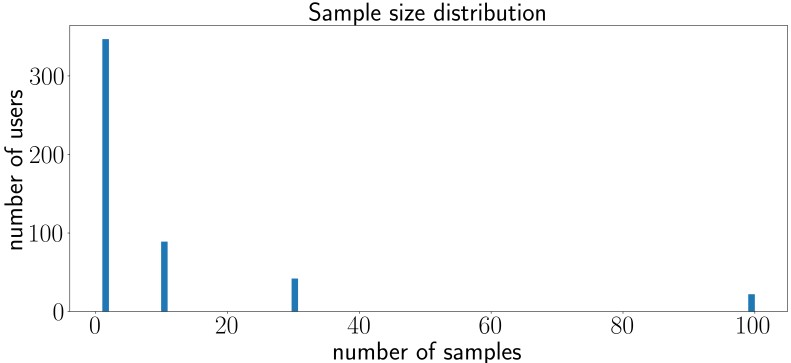

Figure 9: The sample size distribution in the training set across clients.

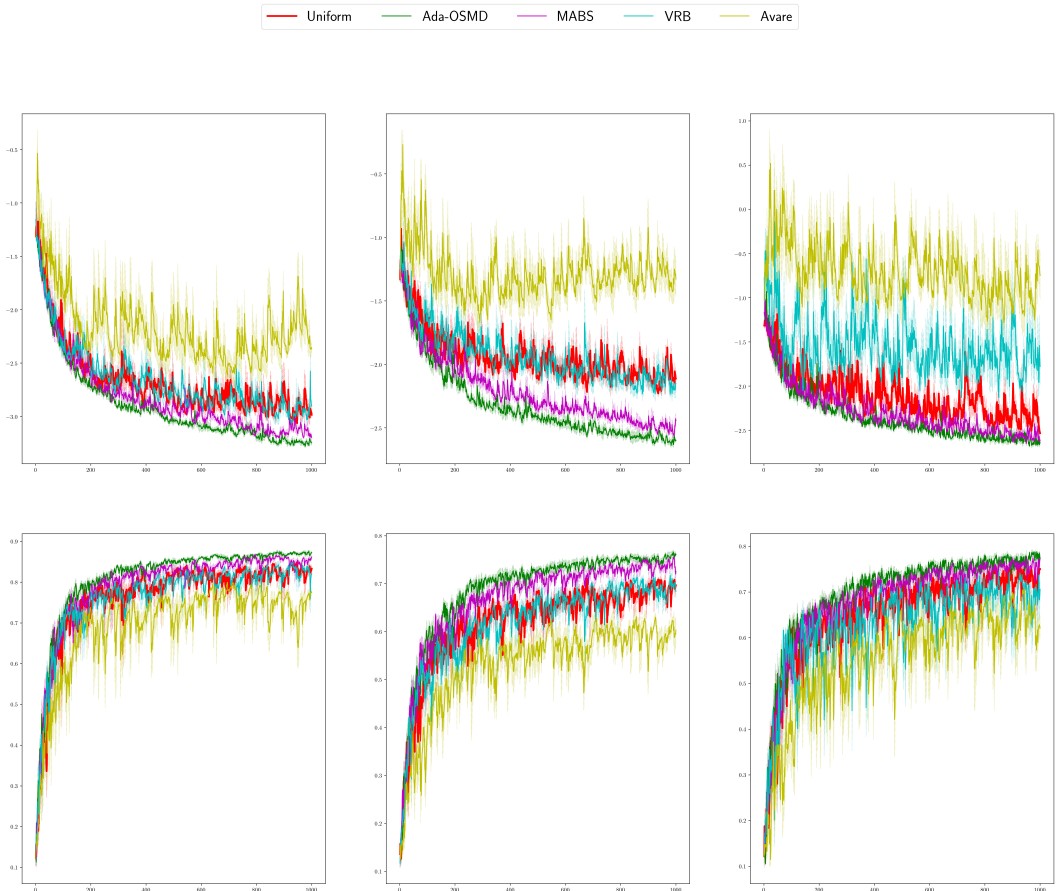

Figure 10: Comparison between Adaptive-OSMD Sampler, uniform sampler and other online learning samplers on real data in terms of training loss (top row) and validation accuracy (bottom row). Different columns correspond to different data sets. The solid line represents the mean and the shadow area represents mean $\pm\, 0.5 \times$ standard error. Adaptive-OSMD Sampler is both faster and more stable. The result is the average performance over 5 independent runs.

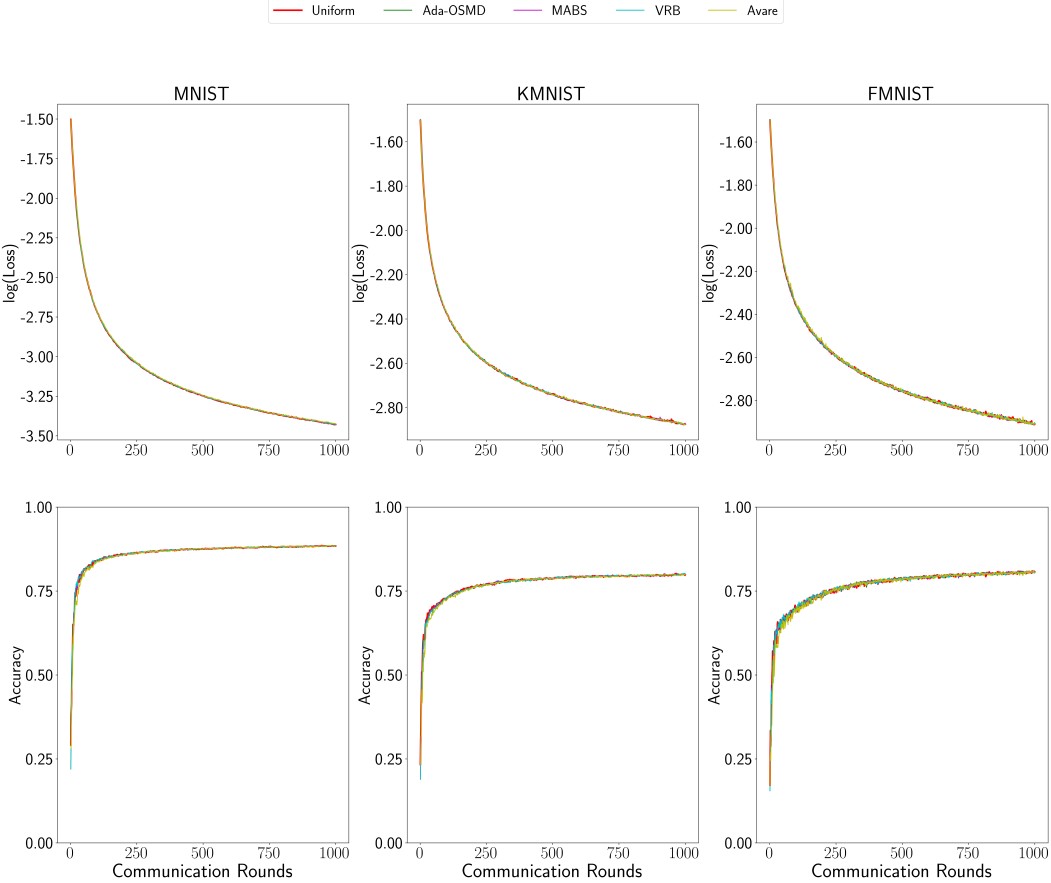

Figure 11: Comparison between Adaptive-OSMD Sampler, uniform sampler and other online learning samplers on real data in terms of training loss (top row) and validation accuracy (bottom row) under balanced samples size. Different columns correspond to different data sets. We see that all methods perform similarly.

demonstrated the benefits of our approach over both widely used uniform sampling and other competitors.

In this paper, we have focused on sampling with replacement. However, sampling without replacement would ideally be a more efficient approach. In Section C.5, we discussed a natural extension of Adaptive-OSMD Sampler to a setting where sampling without replacement is used. However, this approach does not directly minimize the variance of the gradient $g^t$. When sampling without replacement is used, the variance function becomes more complicated and the design of an algorithm to directly minimize the variance is an interesting future direction.

Besides, in federated learning, privacy is a major concern. In this paper, the non-uniform sampling distribution may make the protection of clients' privacy more challenging than uniform sampling. One possible solution is to add noise to the gradient feedback and protect the clients' privacy under the Differential Privacy (DP) concept (Dwork, 2008). However, the added noise may hurt the performance of our sampling design and increase the regret. Studying the trade-off between privacy protection and regret is an important direction for addressing societal concerns in real-world application.

Other fruitful future directions include the design of sampling algorithms for minimizing personalized FL objectives and sampling with physical constraint in FL system, which we discuss next.

### H.1 CLIENT SAMPLING WITH PERSONALIZED FL OBJECTIVE

Data distributions across clients are often heterogeneous. Personalized FL has emerged as one effective way to handle such heterogeneity (Kulkarni et al., 2020). Hanzely et al. (2021) illustrated how many existing approaches to personalization can be studied through a unified framework, and, in this section, we discuss a natural extension of Adaptive-OSMD Sampler to this personalized objective. Specifically, we study the following optimization problem

$$\min_{w,\beta} F(w,\beta) := \sum_{m=1}^{M} \lambda_m \phi(w, \beta_m; \mathcal{D}_m), \tag{45}$$

where $w \in \mathbb{R}^{d_0}$ corresponds to the shared parameter and $\beta = (\beta_1, \ldots, \beta_M)$ with $\beta_m \in \mathbb{R}^{d_m}$ corresponds to the local parameters. The objective in equation 45 coves a wide range of personalized federated learning problems (Hanzely et al., 2021). We further generalize the approach and study the following bilevel optimization problem:

$$\min_{w} h(w) := \sum_{m=1}^{M} \lambda_m F_m(w, \hat{\beta}_m(w)) := \sum_{m=1}^{M} \lambda_m \phi(w, \hat{\beta}_m(w); \mathcal{D}_m)$$
$$\text{subject to} \quad \hat{\beta}_m(w) = \arg\min_{\beta_m} G_m(w, \beta_m) := \phi(w, \beta_m; \bar{\mathcal{D}}_m). \tag{46}$$

When $\mathcal{D}_m = \bar{\mathcal{D}}_m$, then equation 46 recovers equation 45. When $\bar{\mathcal{D}}_m \neq \mathcal{D}_m$, we then optimize the shared and local parameters on different datasets, which may prevent overfitting. The formulation in equation 46 is closely related to the implicit MAML (Rajeswaran et al., 2019).

In the following, we use $\nabla_w$ to denote a partial derivative with respect to $w$ with $\beta_m$ fixed, $\nabla_{\beta_m}$ to denote a partial derivative with respect to $\beta_m$ with $w$ fixed, and $\nabla$ to denote a derivative with respect to $w$ where $\beta_m(w)$ is treated as a function of $w$. Let $\nabla^2_{\beta_m \beta_m} G_m(w, \beta_m) \in \mathbb{R}^{d_m \times d_m}$ be the Hessian matrix of $G_m$ with respect to $\beta_m$ where $w$ is fixed, and $\nabla^2_{w\beta_m} G_m(w, \beta_m) \in \mathbb{R}^{d_0 \times d_m}$ be the Hessian matrix of $G_m$ with respect to $w$ and $\beta_m$, that is,

$$\left[\nabla^2_{\beta_m \beta_m} G_m(w, \beta_m)\right]_{i,j} = \frac{\partial G_m(w, \beta_m)}{\partial \beta_{m,i} \beta_{m,j}} \quad \text{for all } i, j = 1, 2, \ldots, d_m,$$

$$\left[\nabla^2_{w\beta_m} G_m(w, \beta_m)\right]_{i,j} = \frac{\partial G_m(w, \beta_m)}{\partial w_i \beta_{m,j}} \quad \text{for all } i = 1, 2, \ldots, d_0, j = 1, 2, \ldots, d_m.$$

By the implicit function theorem, we have

$$\nabla h(w) = \underbrace{\frac{1}{M} \sum_{m=1}^{M} \lambda_m \nabla_1 F_m(w, \hat{\beta}_m(w))}_{\nabla_1 h(w)} + \underbrace{\frac{1}{M} \sum_{m=1}^{M} \lambda_m \nabla_2 F_m(w, \hat{\beta}_m(w))}_{\nabla_2 h(w)} \tag{47}$$

where

$$\nabla_1 F_m(w, \hat{\beta}_m(w)) := \nabla_w F_m(w, \hat{\beta}_m(w)),$$

$$\nabla_2 F_m(w, \hat{\beta}_m(w)) := -\nabla^2_{w\beta_m} F_m(w, \hat{\beta}_m(w)) \left[\nabla^2_{\beta_m \beta_m} F_m(w, \hat{\beta}_m(w))\right]^{-1} \nabla_{\beta_m} F_m(w, \hat{\beta}_m(w)).$$

There are two parts to $\nabla h(w^t)$ and, therefore, instead of choosing a single subset of clients for computing both parts, we decouple $S^t$ into two subsets $S_1^t$ and $S_2^t$, $S^t = S_1^t \cup S_2^t$. We use clients in $S_1^t$ to compute local updates of the first part, and clients in $S_2^t$ to compute the local updates of the second part. To get an estimate of $\nabla h(w)$, we can estimate $\nabla_1 h(w)$ and $\nabla_2 h(w)$ separably and then combine. Assume that $g_{1,m}^t$ is an estimate of $\nabla_1 F_m(w, \hat{\beta}_m(w))$ and $g_{2,m}^t$ is an estimate of $\nabla_2 F_m(w, \hat{\beta}_m(w))$, we can then construct estimates of $\nabla_1 h(w)$ and $\nabla_2 h(w)$ as

$$g_1^t = \frac{1}{K_1} \sum_{m \in S_1^t} \lambda_m \frac{g_{1,m}^t}{p_{1,m}^t}, \qquad g_2^t = \frac{1}{K_2} \sum_{m \in S_2^t} \lambda_m \frac{g_{2,m}^t}{p_{2,m}^t},$$

where $K_1 = |S_1^t|$ and $K_2 = |S_2^t|$. Then $g^t = g_1^t + g_2^t$ is an estimate of $\nabla h(w)$.

We design $p_1^t$ and $p_2^t$ to choose $S_1^t$ and $S_2^t$ by minimizing the variance of the gradients. Note that

$$\min_{p_1^t} \mathbb{E}_1 \left[ \mathbb{E}_{S_1^t} \left[ \left\| g_1^t - \nabla_1 h(w^t) \right\|^2 \right] \right] + \min_{p_2^t} \mathbb{E}_2 \left[ \mathbb{E}_{S_2^t} \left[ \left\| g_2^t - \nabla_2 h(w^t) \right\|^2 \right] \right]$$
$$\leq \min_{p_1^t = p_2^t = p^t} \mathbb{E} \left[ \mathbb{E}_{S^t} \left[ \left\| g_1^t - \nabla_1 h(w^t) + g_2^t - \nabla_2 h(w^t) \right\|^2 \right] \right],$$

so that the decomposition allows us to better minimize the variance. We term this approach as *doubly variance reduction for personalized Federated Learning*. The first part minimizes the variance of updates to the shared global parameter, when the best local parameters are fixed; and the second part minimizes the variance of updates to local parameters, when the global part is fixed. While these two parts are related, any given machine will have different contributions to these two tasks.

Adaptive-OSMD Sampler can be used to minimize the variance for both parts of the gradient. We note that this is a heuristic approach to solving the client sampling problem when minimizing a personalized FL objective. Personalized FL objectives have additional structures that should be used to design more efficient sampling strategies. Furthermore, designing sampling strategies that improve the statistical performance of trained models, rather than improving computational speed, is important in the heterogeneous setting. Addressing these questions is an important area for future research.

## H.2 SAMPLING WITH PHYSICAL CONSTRAINT IN FL SYSTEM

In this paper, we assume that all clients are available in each round. However, in practical FL applications, a subset of the clients may be inactive due to physical constraints, thus we have to assign zero probabilities to them. In this section, we propose a simple extension of our proposed sampling method to such case.

Specifically, denote the subset of clients that are active at the beginning of round $t$ as $I^t \subseteq [M]$. If we have $|I^t| \leq K$, we can then use all clients in $I^t$ to make updates in round $t$; otherwise, we would like to choose a smaller subset $S^t \subseteq I^t$ to participate. This can be achieved by rescaling the output sampling distribution of any of our proposed methods, which we denote as $\hat{p}^t$. We let $\tilde{p}_m^t = \hat{p}_m^t / (\sum_{i \in I^t} \hat{p}_i^t)$ and $\tilde{p}_m^t = 0$ for all $m \notin I^t$. We can then use $\tilde{p}_m^t$ to choose $S^t$ from $I^t$.

However, analyzing such a method in terms of convergence and regret guarantee is highly non-trivial. Typically, for general active clients sequence $\{I^t\}_{t=1}^T$, the optimization algorithms are not guaranteed to converge even if we involve all clients in $I^t$ in each round. This can happen, for example, if a client is active for only once in the whole training process. Thus, to ensure convergence, we need additional assumptions about $\{I^t\}_{t=1}^T$. Moreover, deriving regret bound is also very challenging, as assigning zero probability to any client will make the variance-reduction loss unbounded and thus the regret can be arbitrarily large. To achieve such theoretical result, one may need to appropriately redefine the regret concept. Such an analysis is beyond the scope of this paper and we leave it for future research.

