# OpenReview forum: "Adaptive Client Sampling in Federated Learning via Online Learning with Bandit Feedback"
_ICLR.cc/2023/Conference — Submitted to ICLR 2023_

### Official Review · Reviewer_dGNg · 2022-10-20

**Confidence:** 2
**Correctness:** 4
**Technical Novelty And Significance:** 2
**Empirical Novelty And Significance:** 2
**Recommendation:** 5

**Clarity, Quality, Novelty And Reproducibility:**

Clarity: the organization of this paper is a little strange, which should be improved.

Quality: see strengths and weaknesses.

Novelty: the studied problem is not new, but the proposed method and the dynamic regret bound are new to me.

Reproducibility: the proposed algorithm and the analysis are presented sufficiently. Moreover, the code is also provided.

**Strength And Weaknesses:**

#Strength
1) Previous studies about online sampling variance minimization with bandit feedback focus on static regret. By contrast, this paper considers dynamic regret. To achieve a dynamic regret bound, the authors propose the online stochastic mirror descent (OSMD) algorithm, which is the main contribution of this paper and is new to me.
2) The combination of the OSMD sampler with federated optimization seems to be reasonable and the convergence rate has also been proved.

#Weakness
1) The organization of this paper is a little strange. In the main paper, the authors do not provide experimental results, which seems to be important for studies on federated learning. Moreover, it would be better if the complete related work is introduced in the main paper.
2) The experimental results provided in the appendix are not sufficient to verify the advantage of the proposed method. First, from Figures 2 and 10, the performance of the existing MABS method is close to that of the proposed method in terms of loss. Second, the authors do not provide experiments to show the necessity of dynamic regret, which is the main novelty of the paper.
3) I agree that the proposed OSMD is applicable beyond federated learning. Therefore, it is a little strange why the authors do not directly study the larger problem, namely stochastic optimization. Moreover, it seems that federated learning does not bring an essential challenge compared with stochastic optimization.

**Summary Of The Paper:**

This paper studies adaptive client sampling in federated optimization and formulates it as the problem of online sampling variance minimization with bandit feedback. To address this problem, the authors propose an online stochastic mirror descent (OSMD) algorithm and establish dynamic regret bound. Moreover, the convergence rate of federated optimization with the proposed OSMD sampler is also established.

**Summary Of The Review:**

By considering the issue of insufficient experiments, I tend to reject this paper.

---

### Official Review · Reviewer_cLap · 2022-10-25

**Confidence:** 3
**Correctness:** 3
**Technical Novelty And Significance:** 3
**Empirical Novelty And Significance:** 2
**Recommendation:** 5

**Clarity, Quality, Novelty And Reproducibility:**

- The paper is not organized well. There lacks a balance of theory and practice in the main paper, as the experimental results are critical but they are left in the appendix entirely. Related works are presented but with an arbitrarily chosen topic in the main paper while the others in appendix. Instead, the authors spend some space discussing the organizing of the main paper, which is wasteful of the space. The overall organization of the paper is poor.

**Strength And Weaknesses:**

+ Formulating the client selection problem as an online learning one with bandit feedback is an interesting idea, which allows the use of dynamic regret to measure the performance.
+ Both convergence and regret are analyzed.

Weakness:
- The empirical performance of the proposed adaptive client selection is not well presented. The authors are suggested to report experimental results with standard FL tasks for both IID and non-IID clients on (FE)MNIST, CIFAR, and Shakespeare datasets.
- From a pure theoretical point of view, there is limited novelty of this work (especially from the online learning aspect). The theoretical results and their derivations are fairly standard. The gradient in (7) follows the same idea as the unbiased estimate of $l_t(q)$, i.e., the expectation of number of times client $m$ is selected equals $Kp_m^t$, and I do not understand why this is novel.
- I'm wondering why the authors did not choose adversarial learning as the tool to study this FL client selection problem.
- Sampling with replacement is used in the main paper. It is unclear how this is relevant in practice -- it would lead to potentially selecting the same client multiple times in one round of FL, which is unrealistic.


----
Post-rebuttal comment:
I thank the authors for their responses, in particular providing new simulation results for FL. I would encourage the authors to carefully consider the organization of this paper -- if the theoretical results of online learning are not the main contributions, maybe the authors can consider moving the theoretical part to appendix, while moving the experimental results to the main paper.

**Summary Of The Paper:**

This paper studies client selection in federated learning. The authors cast this problem as an online learning task with bandit feedback. They proposed to adopt online stochastic mirror descent (OSMD) to minimize the sampling variance.

**Summary Of The Review:**

This work is technically sound and solves an interesting problem of adaptive client selection in federated learning. The solution has solid theoretical foundation. It would be great if the weakness aspects can be resolved.

---

### Official Review · Reviewer_ZCkp · 2022-10-31

**Confidence:** 2
**Correctness:** 4
**Technical Novelty And Significance:** 3
**Empirical Novelty And Significance:** 3
**Recommendation:** 6

**Clarity, Quality, Novelty And Reproducibility:**

I am not an expert in the area of federated learning. But it is interesting to see online learning applied to client sampling with theoretical guarantee, which seems novel to me.

**Details Of Ethics Concerns:**

I do not find any concerns.

**Strength And Weaknesses:**

Strength
* This study establishes the regret upper bound for the first time.
* Numerous tests are performed to validate the suggested algorithm.

Weaknesses
* The presentation of this paper could be improved. It could be preferable if the author(s) included Section 3, which applies the suggested OSMD sampler to MINI-BATCH SGD, in the appendix rather of the main paper and included the experiments there instead. Additionally, there is no concluding paragraph.

**Summary Of The Paper:**

This work examines adaptive client sampling in federated learning. The paper's major contribution is to treat client sampling as an online learning problem and to create an algorithm employing a novel sampler (OSMD) with a nice theoretical guarantee.

**Summary Of The Review:**

This paper established the first regret bound of applying online learning technique on the client sampling in federated sampling. But in my opinion, the presentation could be further improved. Furthermore, it appears to me that a related sampler has already been described in Borsos et al., 2018, so it would be better if the author(s) could describe the technical challenges involved in establishing the necessary bound. Overall I suggest weak accept.

---

### Official Review · Reviewer_Hgfz · 2022-11-04

**Confidence:** 3
**Clarity, Quality, Novelty And Reproducibility:** This paper is well-written in general.
**Correctness:** 4
**Technical Novelty And Significance:** 3
**Empirical Novelty And Significance:** 2
**Recommendation:** 8

**Strength And Weaknesses:**

Strength:
1. Although I didn't check all the proof, the results in this paper look intuitive and technically sound.
2. This paper is well-motivated, and the proposed OSMD method is of interests to federated learning and stochastic optimization.

Weakness & Questions:
1.  There lacks enough discussions and comparison with federated learning algorithms, e.g. a table summarizing the settings and theoretical results may be helpful. Currently, the discussions in this paper is more about variance reduction for gradient estimation in stochastic optimization algorithm. It is a bit unclear to me the contribution/improvement compared with existing federated learning algrithms. I would appreciate it if the authors could provide more explanations. For example, can the proposed method help attain linear convergence to the global optimum for strongly convex and smooth functions, as in (Mitra et. al. 2021)?

Mitra, A., Jaafar, R., Pappas, G.J. and Hassani, H., 2021. Linear convergence in federated learning: Tackling client heterogeneity and sparse gradients. Advances in Neural Information Processing Systems, 34, pp.14606-14619.

2. I am curious, in Algorithm 3, is it possible to apply OSMD to learn the sampling probabilities of mini-batch SGD in line 8, and will it be helpful?

3. At the end of section 2.1, the authors mentioned "will treat the environment as deterministic". Can the authors elaborate more on this? What is our assumption on the sequence of variance reduction loss function, can it be generated by an adaptive adversary?

**Summary Of The Paper:**

Stochastic optimization algorithms typically work by estimating the gradient of the cost function on the fly, and thus, their convergence rate depends on the cumulative variance of the gradient estimation. Therefore, proper selection of the sampling probability on the data points (that are used to compute the gradient) can help reduce this variance.
Similarly, under the federated/distributed setting, the server estimates the gradient by sampling a subset of clients, and this paper proposed an online stochastic mirror descent algorithm to decide the sampling probabiilty on the clients. As the optimal sampling probability changes in every round, the author proved the dynamic regret of online stochastic mirror descent algorithm.

**Summary Of The Review:**

This paper is technically sound, and the idea of using OSMD for client sampling is natural and well-motivated. Therefore, I recommend accept.

---

### Decision · Program_Chairs · 2023-01-20

**Decision:**

Reject

**Justification For Why Not Higher Score:**

Reviewers share remaining concerns on limited theoretical contribution, theoretical result (dynamic regret bound) not verified by experiments, and paper organization should be improved.

**Justification For Why Not Lower Score:**

N/A

**Metareview: Summary, Strengths And Weaknesses:**

This paper studied client selection in federated learning, formulated as an online learning task with bandit feedback. The authors proposed adaptive client sampling by applying an online stochastic mirror descent (OSMD) algorithm to minimize the sampling variance. Dynamic regret bound and convergence rate of proposed method are analyzed.

All reviewers appreciate that the idea of connecting client selection with online learning is well-motivated and interesting. Theoretical results also look solid to reviewers. However, there are several shared concerns: 1. Theoretical contribution is not verified by experiments. Since the authors proved dynamic regret and argued advantage of it, experiments (at least simulations) should be conducted in a dynamic environment.  2. Current organization of the paper should be improved. The authors are suggested to move the experimental results from appendix to the main paper, and move some theoretical discussion to appendix under space constraint, which requires substantial revision. The authors are encouraged to resolve them in the next version and resubmit the paper in future venues.

**Summary Of Ac-Reviewer Meeting:**

Reviewers shared their thoughts on remaining concerns and some of them are not addressed in rebuttal. All reviewers appreciate that the idea of connecting client selection with online learning is well-motivated and interesting.  The mentioned concerns include:

1. Theoretical contribution is limited and is not verified by experiments. Since the authors proved dynamic regret instead of static regret, experiments (at least simulations) should be conducted in a dynamic environment. Currently results are all in static environment. Reviewer dGNg mentioned this is a remaining concern and other reviewers agreed with it.

2. Current organization of the paper, where experiments are all in appendix, should be greatly improved. Reviewers dGNg and cLap suggest that the authors should move the experimental results from appendix to the main paper, and move some theoretical discussion to appendix under space constraint. The paper is likely to be more interesting to the readers if focusing on the proposed algorithm and experimental validation, instead of the theory part. This requires substainal revision of the paper.

Reviewer Hgfz, who was the most positive reviewer of the paper, agreed that the paper should be further improved given the remaining concerns. A consensus is reached in recommending rejection while encouraging resubmssion to future venues.